# UBR5 promotes antiviral immunity by disengaging the transcriptional brake on RIG-I like receptors

Duomeng Yang [1] ✉, Tingting Geng[1], Andrew G. Harrison [1], Jason G. Cahoon[1], Jian Xing[2], Baihai Jiao[3], Mark Wang[1], Chao Cheng [4], Robert E. Hill [5], Huadong Wang[6], Anthony T. Vella [1], Gong Cheng [7], Yanlin Wang[3] & Penghua Wang [1] ✉

The Retinoic acid-Inducible Gene I (RIG-I) like receptors (RLRs) are the major viral RNA sensors essential for the initiation of antiviral immune responses. RLRs are subjected to stringent transcriptional and posttranslational regulations, of which ubiquitination is one of the most important. However, the role of ubiquitination in RLR transcription is unknown. Here, we screen 375 definite ubiquitin ligase knockout cell lines and identify Ubiquitin Protein Ligase E3 Component N-Recognin 5 (UBR5) as a positive regulator of RLR transcription. UBR5 deficiency reduces antiviral immune responses to RNA viruses, while increases viral replication in primary cells and mice. *Ubr5* knockout mice are more susceptible to lethal RNA virus infection than wild type littermates. Mechanistically, UBR5 mediates the Lysine 63-linked ubiquitination of Tripartite Motif Protein 28 (TRIM28), an epigenetic repressor of RLRs. This modification prevents intramolecular SUMOylation of TRIM28, thus disengages the TRIM28-imposed brake on RLR transcription. In sum, UBR5 enables rapid upregulation of RLR expression to boost antiviral immune responses by ubiquitinating and de-SUMOylating TRIM28.

When invaded by a virus, a host produces a rapid innate immune response initiated by pathogen pattern recognition receptors (PRRs), including Toll-like receptors (TLRs), retinoic acid-inducible gene I (RIG-I) like receptors (RLRs), the cyclic GMP-AMP (cGAMP) synthase (cGAS), and nucleotide-binding oligomerization domain (NOD)-like receptors (NLRs). These signaling pathways can initiate rapid interferon (IFN) and inflammatory responses that are essential for controlling viral replication. The relative contribution of each PRR to innate antiviral immunity varies with each viral species and tissue cell type. Of note, RIG-I and melanoma differentiation-associated protein 5 (MDA5) are

ubiquitous and the primary viral RNA sensors. Once engaged by viral RNA, RLRs undergo Lysine (K) 63-linked polyubiquitination, oligomerize, and interact with Mitochondrial Antiviral Signaling Protein (MAVS), which ignites a signaling event, leading to transcription of immune genes, in particular type I/III IFNs[1]. RIG-I senses short 5'-pp or ppp-RNA and is essential for the initiation of immune responses to *Orthomyxoviridae* such as influenza and *Rhabdoviridae* including Vesicular Stomatitis Virus (VSV). MDA5 prefers long dsRNA and is essential for the induction of antiviral immunity to *Picornaviridae* such as Encephalomyocarditis Virus (EMCV) and *Coronaviridae* like Severe

[1]Department of Immunology, School of Medicine, UConn Health, Farmington, CT 06030, USA. [2]Department of Neuroscience, School of Medicine, UConn Health, Farmington, CT 06030, USA. [3]Department of Medicine, School of Medicine, UConn Health, Farmington, CT 06030, USA. [4]Department of Medicine, Baylor College of Medicine, Houston, TX 77030, USA. [5]MRC Human Genetics Unit, Institute of Genetics and Molecular Medicine at the University of Edinburgh, Western General Hospital, Edinburgh, EH4 2XU, UK. [6]Department of Pathophysiology, Key Laboratory of State Administration of Traditional Chinese Medicine of the People's Republic of China, School of Medicine, Jinan University, Guangzhou 510632 Guangdong, China. [7]Tsinghua University-Peking University Joint Center for Life Sciences, School of Medicine, Tsinghua University, 100084 Beijing, China. ✉e-mail: dyang@uchc.edu; pewang@uchc.edu

Acute Respiratory Syndrome Coronavirus 2 (SARS-CoV-2)[2,3]. RIG-I exists in an auto-inhibited state in the absence of viral RNA ligands, with its two N-terminal caspase activation and recruitment domains (CARDs) masked by the C-terminal ligand binding domain, while the CARDs of MDA5 are constitutively exposed. Thus, MDA5 signaling could be activated even in the absence of ligands and contribute to pathogenesis of autoimmune diseases[3]. Indeed, an A946T variant is irresponsive to EMCV infection but is constitutively active and is associated with systemic lupus erythematosus (SLE)[4,5].

To minimize auto-activation, the basal expression level of RLRs is generally very low. Upon viral infection, RLR expression is rapidly upregulated in order to amplify and sustain antiviral immune responses[6]. Notably, type I/III IFNs are potent inducers of RLR transcription via the Janus kinase (JAK)- signal transducer and activator of transcription (STAT1/2) signaling transduction pathway[7,8], serving as positive feedback to RLR signaling. MDA5 expression can be also induced by a cytokine-independent manner during some viral infections[9]. In addition to transcriptional regulation, RLR signaling is tightly regulated by posttranslational modifications, among which ubiquitination is one of the best characterized and most important. Although many ubiquitin ligases (E3) have been identified to either positively or negatively regulate RLRs post-translationally[1], the E3 ligases directly involved in RLR transcription are largely unknown. To this end, we attempt to identify new E3 ligases critical for RLR signaling by a systemic approach. We have performed an unbiased screening of 375 individual ubiquitin E3 ligase knockout cell lines and identified Ubiquitin Protein Ligase E3 Component N-Recognin 5 (UBR5) as a positive regulator of RLR transcription. UBR5 is a large (~309.4 kDα) and highly conserved protein in metazoans (murine and human proteins are 98% identical). It may mediate K48-linked ubiquitination and degradation of many proteins that function in DNA damage response, metabolism, transcription, and apoptosis, according to the N-end rule[10]. Here we show that UBR5 mediates K63-linked ubiquitination of Tripartite Motif Protein 28 (TRIM28), an epigenetic repressor of RLRs. TRIM28 binds genomic DNA via Krüppel-Associated Box and Zinc Finger (KRAB-ZNF)-containing transcription factors, undergoes intramolecular modification by Small Ubiquitin-like Modifiers (SUMOylation), then recruits the histone methylation machinery to facilitate chromatin compaction[11]. We demonstrate that UBR5-mediated ubiquitination of TRIM28 prevents its intramolecular SUMOylation and relaxes RLR promoter chromatin. This mechanism identifies a new point of intervention for viral control.

## Results

### Construction of stable E3 knockout cell lines

The human genome encodes ~617 putative ubiquitin E3 ligases and accessory genes based on the presence of signature "catalytic" domains, as well as the domains characteristic of substrate-recognition subunits of multi-subunit RING (Really Interesting New Gene) finger (RNF)-dependent E3[12]. However, only ~377, predominantly the HECT (homologous to E6-associated protein C-terminus), RING, and U-box proteins, are definite E3. To this end, we first attempted to construct an array of individual stable E3 ligase knockouts using a CRISPR-Cas9 method in 2fTGH (a human lung fibroblast cell line) with an ISRE (interferon-stimulated response element)-driven firefly luciferase reporter (2fTGH-ISRE-Luc). We picked a pre-designed gene-unique guide (g) RNA sequence with maximal on-target and/or minimal off-target efficiency for each gene from a commercial source (Integrated DNA Technologies). We cloned each gRNA into a lentiCRISPR-v2 vector[13] and confirmed correct insertion by DNA sequencing. We then produced lentiviral particles in HEK293T cells, transduced 2fTGH-ISRE-Luc cells, and selected stably transduced cells with puromycin (Extended Data Fig. 1a). Finally, we succeeded constructing 375 individual lentiviral

vectors (Supplementary Data 1) but failed to recover 7 knockout lines after antibiotic selection. Loss of these individual genes per se could be lethal or render cells highly sensitive to antibiotic stress. To estimate the on-target and knockout efficiency of these guide RNA, we randomly picked nine knockouts and validated by a T7 endonuclease I (T7EI) mismatch cleavage assay. This method is cost-effective and fast, though it generally underestimates the on-target efficiency of a gRNA when compared to next generation sequencing (NGS); a ~60% editing efficiency detected by this assay is equivalent to ~95% efficiency by NGS[14]. Eight out of nine gRNA presented >60% editing efficiency by T7EI (Extended Data Fig. 1b, c).

### Identification of UBR5 as a positive regulator of the RLR pathways

By employing the E3 knockout 2fTGH-ISRE-Luc library, we attempted to identify E3 regulators of the RLR and cGAS pathways and type I IFN receptor signaling via JAK-STAT1/2 [Janus kinase (JAK)-signal transducer and activator of transcription (STAT)]. The ISRE-Luc reporter is designed for monitoring the activity of the JAK-STAT1/2 signaling pathway induced by type I/III IFNs, thus also reflective of the functionality of the PRR pathways. We stimulated cells with a high molecular weight polyinosinic-polycytidylic acid [poly (I:C)] (a MDA5 agonist), IFN-stimulatory DNA (ISD, a cGAS agonist), and recombinant human IFN-β (an IFN receptor-JAK-STAT1/2 agonist) (Fig. 1a). We optimized the concentration and time of stimulation of each ligand (Extended Data Fig. 2), and included wild type (WT, empty lentiviral vector), STING$^{-/-}$, MAVS$^{-/-}$, and IFNAR1$^{-/-}$ as positive controls for ISD, poly (I:C), and IFN-β stimulation, respectively, in each batch of screening (~20 knockout lines) (Extended Data Fig. 1d). Many known E3 ligases regulating the RLR-MAVS, cGAS-STING and JAK-STAT1/2 were validated[1] (Fig. 1b−d, Supplementary Data 2). Of note, the poly (I:C)-stimulated ISRE-Luc activity was most dramatically reduced (~10-fold) in UBR5$^{-/-}$ (highlighted in dark red) among all the E3 knockouts including TRIM25 and TRIM65, known E3 ligases for MDA5[1,15], suggesting that UBR5 positively regulates MDA5 signaling (Fig. 1e). We observed that ISD-stimulated ISRE activity was moderately downregulated in UBR5$^{-/-}$ cells (~40%) (Fig. 1f), suggesting that UBR5 might play a minor role in the cGAS pathway. However, IFN-β-induced ISRE was not overtly affected (Fig. 1g). These results prompted us to focus on the role of UBR5 in RLR signaling.

### UBR5 is critical for RLR signaling

To validate the screening results, we constructed additional, independent UBR5 knockout lines with unique gRNA #2 and #3. These gRNAs showed variable knockout efficiency, with gRNA #1 (used in the initial screening) being the best, then #2 and #3 (Fig. 2a). Of note, the knockout efficiency was well correlated with the level of reduction in ISRE-Luc activity, secreted type I IFN protein and IFNB1 mRNA expression level in the UBR5$^{-/-}$ cells (2fTGH) following poly (I:C) treatment (Fig. 2b−d). The gRNA #1 almost completely abolished UBR5 protein expression and was correlated with the greatest reduction in the type I IFN expression. We next reproduced the UBR5$^{-/-}$ (gRNA #1) phenotype with a time course (Fig. 2e, f), and in parallel with IFIH1$^{-/-}$ (encoding MDA5) in another cell line HEK293T (Fig. 2g, h). Consistently, depletion of UBR5 by siRNA in HEK293T cells reduced the type I IFN expression following poly (I:C) transfection (Fig. 2i, Extended Data Fig. 3g). To further validate the siRNA results, we transfected HEK293T cells with a UBR5 expression plasmid (or empty vector pcDNA 3.1), a plasmid encoding a firefly luciferase under the human IFNB1 promoter, and a plasmid encoding a renilla luciferase under the human beta actin promoter (internal control). Transient overexpression of UBR5 alone or with poly (I:C) in WT cells enhanced IFN-β expression, compared to vector (Fig. 2j, k).

To extend our results to the in vivo setting, we generated an Ubr5 knockout mouse model. Loss of UBR5 is embryonic lethal, and

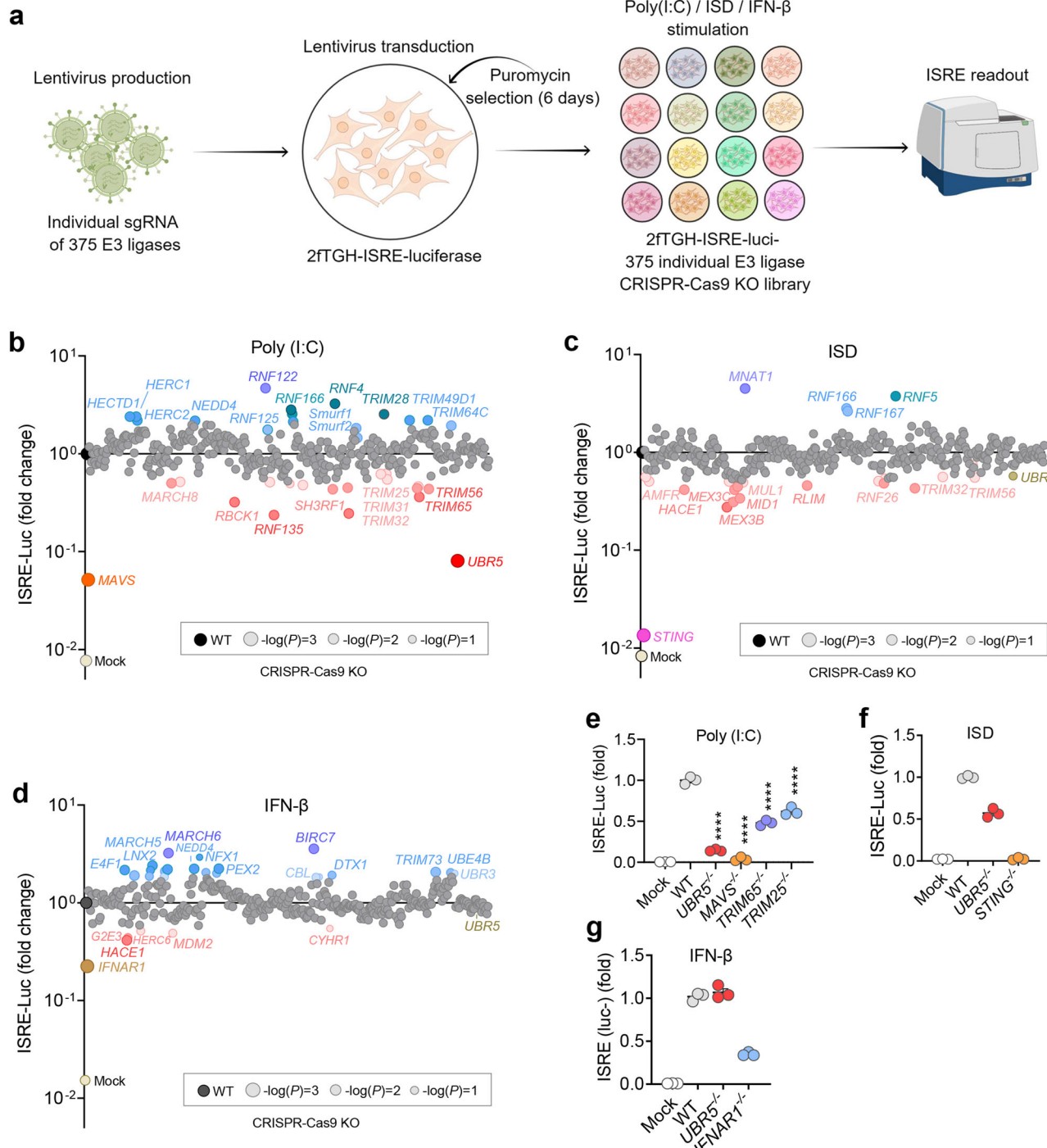

**Fig. 1 | Identification of UBR5 as a positive regulator of MDA5 signaling.**
**a** Graphical illustration of the experimental flow (Created by Figdraw., Figdraw export ID: PUWAU5be43). Individual E3 knockout 2fTGH-ISRE-Luc cells were transfected with (**b**) high molecular weight poly (I:C) to stimulate MDA5, **c** immunostimulatory DNA (ISD) to activate cGAS, or (**d**) treated (no transfection) with recombinant human IFN-β to trigger JAK-STAT1/2 signaling for 12 h. The luciferase activity in each cell line is expressed as the average fold change over WT. The

blue/red/gray dots are the knockouts in which the luciferase activity is higher/lower/ not different, than that in WT (black dot). **e–g** The luciferase results for select E3 knockouts and positive controls. Data are presented as mean ± S.E.M, ordinary one-way ANOVA with Dunnett's test; $n = 3$ biological independent experiments; $p$ values are assigned as -log($P$) in **b–d**; ****$p < 0.0001$ vs WT in **e**. Multiplicity adjusted $p$ values are presented. Mock: lipofectamine in **b, c, e, f** or sterile water in **d, g** only. Source data are provided as a Source Data file.

therefore we engineered an inducible knockout model. The tamoxifen-inducible Cre recombinase model (*ERT2*-Cre) has been successfully applied to our research[16]. An advantage of this system is that it allows for temporal induction of gene deletion. Thus, we crossed *Ubr5*^flox/flox[17] with the *ERT2*-Cre line, and generated *ERT2*-Cre^+/− *Ubr5*^flox/flox mice. To induce global *Ubr5* knockout, 1 mg of tamoxifen (dissolved in corn oil)

was administered to each mouse every other day, totaling 5 doses. This line was designated *Ubr5*^iKO (inducible knockout) to distinguish it from constitutive knockouts. The *ERT2*-Cre^+/− *Ubr5*^flox/flox mice treated with corn oil served as the wild-type control (*Ubr5*^WT). We isolated embryonic fibroblasts (MEF) from *Ubr5*^WT and induced *Ubr5* knockout with 4-OH tamoxifen for 5 days ex vivo. The UBR5 protein deletion in

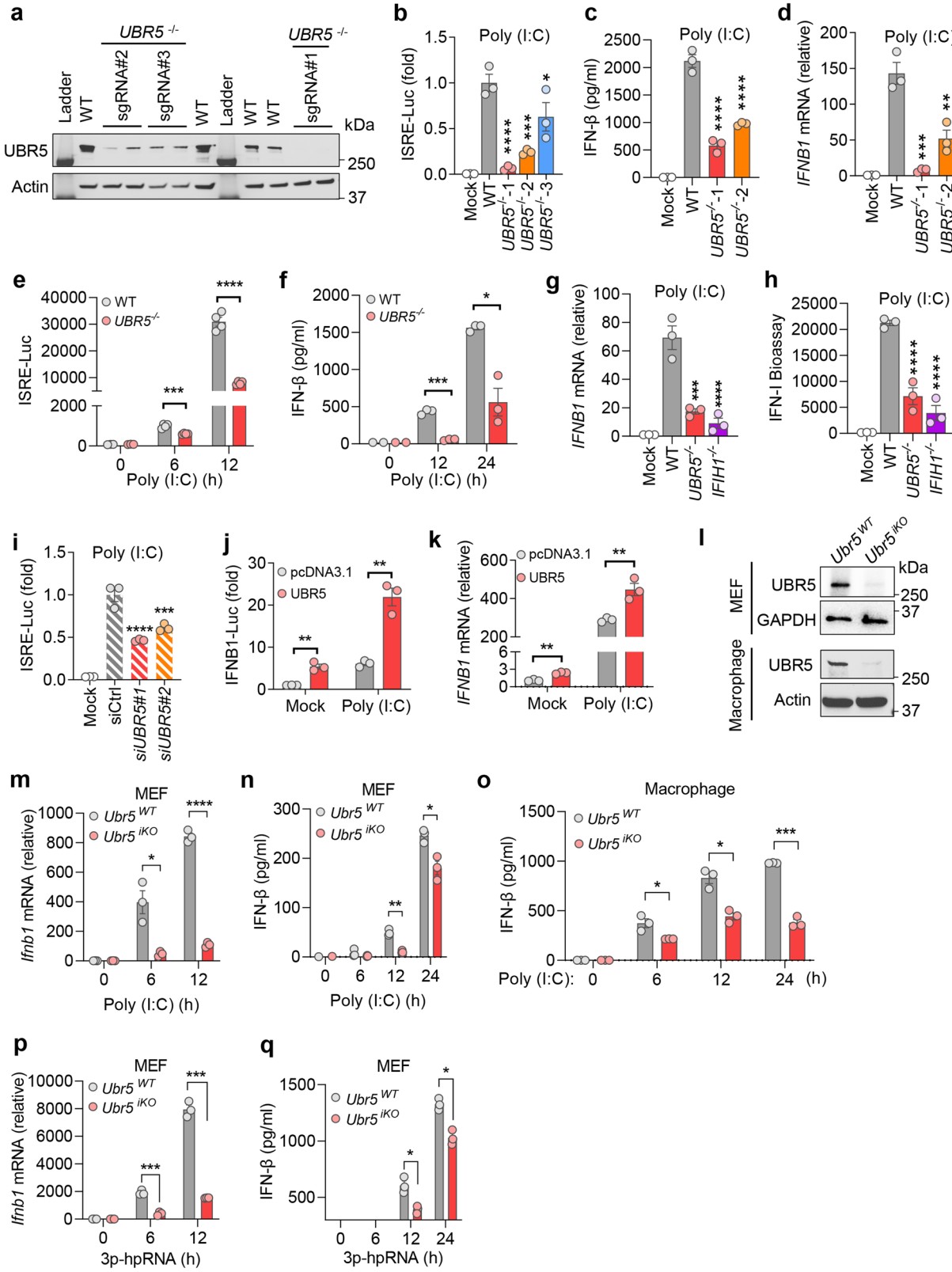

primary bone marrow-derived macrophages (BMDM) and MEFs was confirmed by immunoblotting (Fig. 2l). Consistent with the results from the human cell lines, poly (I:C)-activated *Ifnb1* mRNA and IFN-β protein expression was significantly lower in the *Ubr5*iKO than that in *Ubr5*WT BMDMs and MEFs (Fig. 2m–o). We next asked if UBR5 also regulated RIG-I, and tested the RIG-I ligand, 5′-triphosphated short hairpin RNA (3p-hpRNA). We detected a significant decrease of type I

IFN in MEFs (Fig. 2p, q). However, TLR signaling was largely intact in the *Ubr5*iKO BMDMs (Extended Data Fig. 3), except that poly (I:C)-induced (without transfection) type I IFN expression trended moderately lower without a statistical significance (Extended Data Fig. 3a). These data demonstrated that UBR5 specifically and positively regulates RLRs, and its function is evolutionarily conserved between rodents and humans.

**Fig. 2 | UBR5 is essential for RLR signaling. a** Immunoblots of UBR5 protein in wild type (WT) and three *UBR5*[−/−] (made with unique guide RNA) in 2fTGH-ISRE-Luc cells. Quantification of the (**b**) intracellular ISRE-Luc activity, **c** secreted IFN-β protein by ELISA, **d** cellular *IFNB1* mRNA levels in 2fTGH-ISRE-Luc cells transfected with poly (I:C) for 12 h. Quantification of the (**e**) intracellular ISRE-Luc activity, **f** secreted IFN-β protein, **g** cellular *IFNB1* mRNA, **h** secreted type I IFNs (Bioassay) in WT, *UBR5*[−/−] (made with gRNA #1) and *IFIH1*[−/−] (gene symbol for MDA5) HEK293T cells transfected with poly (I:C) for 12 h. **i** Quantification of ISRE-Luc activity in 2fTGH-ISRE-Luc cells transfected with a negative control siRNA (siCtrl) or UBR5 siRNA for 48 h, then poly (I:C) for 12 h. **j** Measurement of the Luc activity in HEK293T cells transfected with a pcDNA3.1 vector or UBR5 expression plasmid, together with an *IFNB1* promoter-driven firefly luciferase (Luc) and an *ACTIN* promoter-driven renilla Luc plasmid (internal control) for 24 h. The cells were then transfected with poly (I:C) for 12 h. **k** Quantification of the *IFNB1* mRNA in HEK293T cells transfected with a vector or UBR5 expression plasmid for 24 h, then poly (I:C) for 12 h. **l** Immunoblots of UBR5 protein in mouse primary embryonic fibroblasts (MEF) or bone marrow-derived macrophages. Quantification of (**m**) the *Ifnb1* mRNA in MEFs,

secreted IFN-β protein in (**n**) MEFs and (**o**) macrophages transfected with poly (I:C). Quantification of the (**p**) *Ifnb1* mRNA levels, **q** secreted IFN-β protein in MEFs transfected with 5-ppp hpRNA (RIG-I agonist). *Ubr5*[iKO]: *Ubr5* inducible knockout by Tamoxifen. Data presented in **b**–**d**, **g**–**i**: mean ± S.E.M, ordinary one-way ANOVA with Dunnett's test, *n* = 3 biological independent experiments; for **b**: ****$p < 0.0001$, ***$p = 0.0003$, *$p = 0.0305$ vs WT; for **c**: ****$p < 0.0001$ vs WT; for **d**: ***$p = 0.0003$, **$p = 0.0022$ vs WT; for **g**: ***$p = 0.0001$, ****$p < 0.0001$ vs WT; for **h**: ****$p < 0.0001$ vs WT; for **i**: ****$p < 0.0001$, ***$p = 0.0005$ vs siCtrl. Multiplicity adjusted *p* values are presented. Data presented in **e**, **f**, **j**, **k**, **m**–**q**: mean ± S.E.M, two-tailed student's *t* test; for **e**: ***$p = 0.0007$, ****$p < 0.0001$, *n* = 4 biological independent experiments; for **f**: ***$p = 0.0002$, *$p = 0.012$; for **j**: **$p = 0.0015$, **$p = 0.0018$ in sequence; for **k**: **$p = 0.0014$, **$p = 0.0084$ in sequence; for **m**: *$p = 0.0224$, ****$p < 0.0001$; for **n**: **$p = 0.0024$, *$p = 0.0407$; for **o**: *$p = 0.0478$, *$p = 0.0129$, ***$p = 0.0002$ in sequence; for **p**: ***$p = 0.0005$, ***$p = 0.0001$ in sequence; for **q**: *$p = 0.0263$, *$p = 0.0138$ in sequence; *n* = 3 biological independent experiments in **f**, **j**, **k**, **m**–**q**. Adjusted *p* values are presented. Source data are provided as a Source Data file.

## UBR5 is critical for the control of RNA virus infection

We next examined the role of UBR5 in the induction of type I IFN by EMCV that triggers MDA5-depedent immune responses and VSV that activates RIG-I-dependent immune responses, respectively. The intracellular EMCV RNA loads and viral particles in the supernatant of the *Ubr5*[iKO] MEFs were significantly higher than those in the *Ubr5*[WT] cells at 6 and 12 h post infection (*p.i.*) (Fig. 3a, b), while the *Ifnb1* mRNA and IFN-β protein levels were lower in the *Ubr5*[iKO] MEFs (Fig. 3c, d). Next, we employed a reporter VSV with a green fluorescence protein (GFP) integrated into its genome to assess VSV replication. The intracellular VSV glycoprotein (G) level and GFP intensity were noticeably higher (Fig. 3e, f), while the IFN-β protein level was lower, in the *Ubr5*[iKO] MEFs than those in the *Ubr5*[WT] cells (Fig. 3g). We recently showed that SARS-CoV-2 induced-type I IFN response was primarily reliant on MDA5 in a human lung epithelial cell line Calu-3[2]. Thus, we tested if the antiviral function of UBR5 could be extended to SARS-CoV-2. To this end, we generated *UBR5*[−/−], along with positive controls, *MDA5*[−/−], and *MAVS*[−/−] Calu-3 cells by CRSIPR-Cas9 and a lentiviral vector. We first validated each knockout efficiency by immunoblotting and antiviral phenotype with EMCV, then examined SARS-CoV-2 replication (Extended Data Fig. 4). As anticipated, the EMCV RNA loads were increased, while the *IFNB1* and *ISG15* mRNA levels were reduced in all the knockout cells when compared to WT (Extended Data Fig. 4b, c). The SARS-CoV-2 RNA loads and titers were significantly higher in *UBR5*[−/−], as well as in *MDA5*[−/−], and *MAVS*[−/−] than those in WT cells (Extended Data Fig. 4d, e). These results clearly showed an essential role of UBR5 in the control of replication of a broad spectrum of RNA viruses. Next, we asked if this antiviral function of UBR5 is specific to RNA viruses or not. We used a model DNA virus, herpes simplex virus 1 (HSV-1), and found that the HSV-1 DNA load was moderately higher in the *Ubr5*[iKO] cells, even though the *Ifnb1* expression was the same as in the *Ubr5*[WT] cells (Fig. 3h, i), suggesting that UBR5 may interfere with HSV-1 replication independent of type I IFNs.

Having demonstrated the essential role of UBR5 in the control of RNA virus infection in mouse primary cells, we next examined RNA virus pathogenesis in vivo. Immunoblotting confirmed the UBR5 protein deletion in various tissues of the *Ubr5*[iKO] mice (Fig. 4a). These mice succumbed to EMCV infection more rapidly than the *Ubr5*[WT] littermates, regardless of the dose (Fig. 4b, c). EMCV predominantly infected the heart on Day 4 *p.i.* before breaching the blood brain barrier (Extended Data Fig. 5a). The peak viremia and heart viral loads on Day 3 *p.i.* were significantly higher in *Ubr5*[iKO] mice (Fig. 4d–f). Because low doses of EMCV did not induce a robust systemic immune response, we employed a high dose ($1 \times 10^7$ PFU) by intravenous injection (*i.v.*), as described by many labs[18]. The peak type I IFN levels and inflammatory cytokines/chemokines at 8 h *p.i.* were much lower

in the *Ubr5*[iKO] mice than those in the *Ubr5*[WT] littermates (Fig. 4g, h). We observed a similar phenotype of *Ubr5*[iKO] mice infected with VSV, presenting a greater mortality rate and sickness score (Fig. 4i, j). In contrast, the serum cytokine levels during HSV-1 infection were largely normal, though IFN-β and TNF-α trended slightly lower in *Ubr5*[iKO] (Extended Data Fig. 5b).

## UBR5 regulates RLR transcription

Because UBR5 is an established E3 ligase, we asked if UBR5 directly regulates RLR protein function and/or stability by ubiquitination. We examined type I IFN induction by overexpression of RLRs, downstream signaling components including MAVS, TBK1, and IRF3, no differences between WT and *UBR5*[−/−] HEK293T cells were observed (Extended Data Fig. 6a). Of note, the endogenous RIG-I expression was significantly reduced in *UBR5*[−/−] cells upon overexpression of RIG-I-CARD and MAVS (Extended Data Fig. 6b). However, we detected no physical interaction between UBR5 and RLRs by co-immunoprecipitation (Extended Data Fig. 6c, d). These results suggested that UBR5 may not mediate post-translational modification of RLRs and their downstream components, but likely RLR expression.

UBR5 is primarily a nuclear E3 ligase and is known to regulate the stability of transcription factors, enhancers, and repressors[19–21]. Thus, we examined the endogenous RLR protein level in the basal and induced states. UBR5 deficiency led to a substantial reduction in constitutive MDA5 and RIG-I protein levels in primary MEFs and BMDMs, several human cell lines including HEK293T, 2fTGH, Calu-3, and A549, and finally in primary mouse tissues (Fig. 5a, b). However, the MAVS and STING protein levels remained largely unchanged (Extended Data Fig. 7a). The RLR expression increased following poly (I:C) treatment or VSV infection in WT HEK293T cells which remained lower in *UBR5*[−/−] cells (Fig. 5c, d). TBK1 and IRF3 are major components of the RLR pathway, and their phosphorylation was significantly reduced in UBR5 deficient cells upon poly (I:C) stimulation (Fig. 5c). Reconstitution of UBR5 expression in *UBR5*[−/−] cells restored both the basal and poly (I:C)-induced RLR protein expression (Fig. 5e). Overexpression of wild type UBR5 alone enhanced the RLR level, but an E3 ligase deficient *UBR5* mutant (C2768A)[22] failed to do so, suggesting that UBR5 regulates the RLR protein level by ubiquitination (Fig. 5f). The reduction in the RLR protein level in *UBR5*[−/−] could be due to accelerated degradation or deficient transcription. We first performed a cycloheximide chase assay to determine the RLR protein stability. A recombinant FLAG-MDA5 expression plasmid was expressed under the control of a human cytomegalovirus (CMV) promoter in HEK293T cells for 24 h. FLAG-MDA5 protein synthesis was then terminated by cycloheximide and chased for 6 h. The FLAG-MDA5 protein level reduced due to degradation at 6 h after cycloheximide equally in the WT and *UBR5*[−/−]

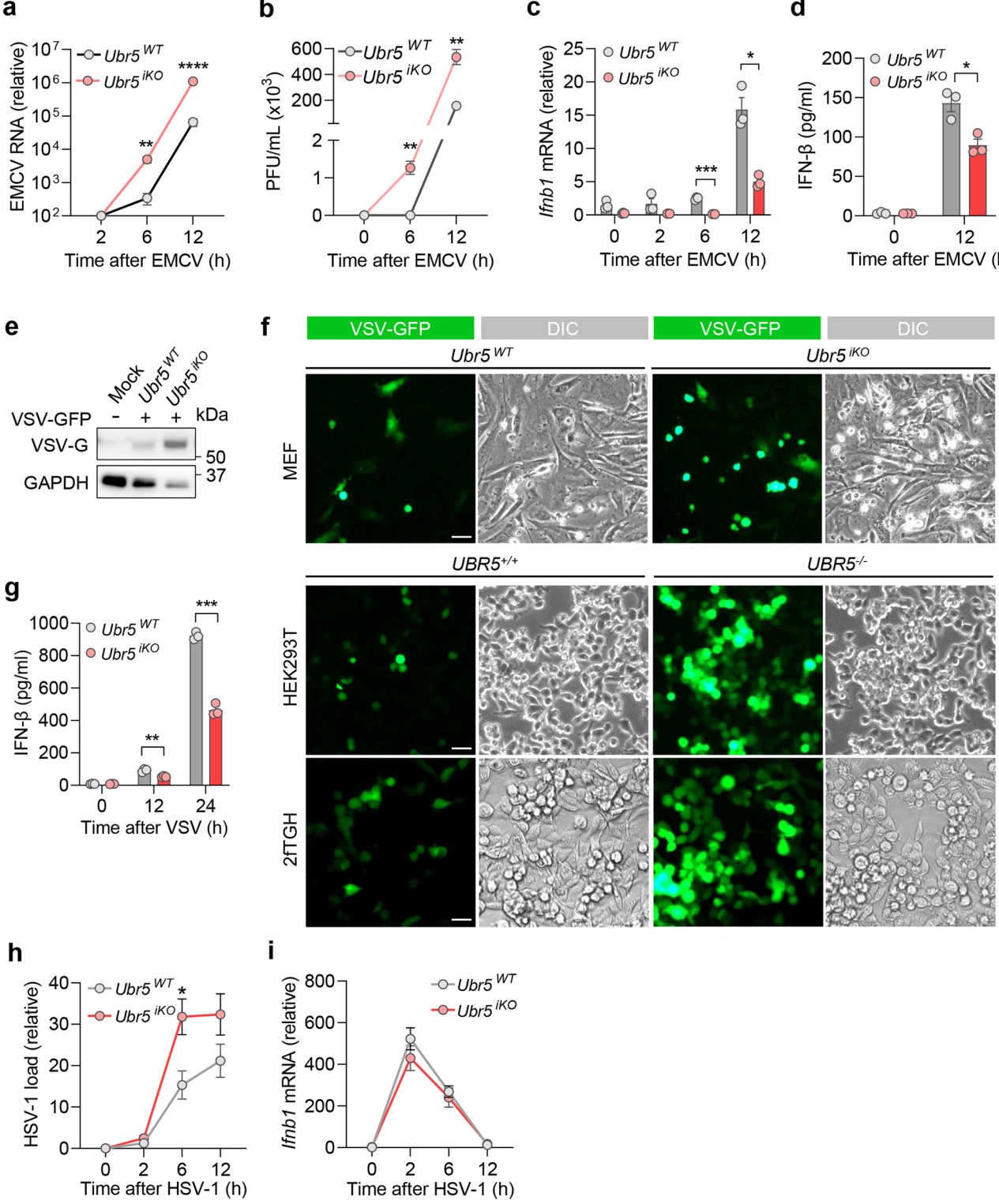

**Fig. 3 | UBR5 is crucial for the induction of type I IFNs by RNA viruses.** Quantification of (**a**) intracellular viral RNA by qRT-PCR, and (**b**) extracellular viral titers (plaque forming units, PFU/mL), in MEFs infected with EMCV at a multiplicity of infection (MOI) of 0.1. Quantification of (**c**) the cellular *Ifnb1* mRNA by qRT-PCR and (**d**) secreted IFN-β by ELISA. **e** The immunoblot of cellular VSV glycoprotein (VSV-G) in MEFs. **f** Fluorescent images of VSV-GFP in three cell types. DIC: differential interference contrast. Scale bar: 50 μM. **g** Quantification of secreted IFN-β protein in

MEFs, infected with VSV-GFP at a MOI of 0.5 for 24 h. Quantification of (**h**) intracellular HSV-1 RNA and (**i**) *Ifnb1* mRNA by qRT-PCR, in MEFs infected with HSV-1 at a MOI of 0.5. Data shown in **a**–**d**, **g**–**i** are presented as mean ± S.E.M, two-tailed Student's *t* test, *n* = 3 biologically independent experiments; for **a**: **$p$ = 0.0038, ****$p$ < 0.0001; for **b**: **$p$ = 0.0040, **$p$ = 0.0040 in sequence; for **c**: ***$p$ = 0.0002, *$p$ = 0.0133; for **d**: *$p$ = 0.0297; for **g**: **$p$ = 0.0075, ***$p$ = 0.0002; for **h**: *$p$ = 0.0299. Adjusted $p$ values are presented. Source data are provided as a Source Data file.

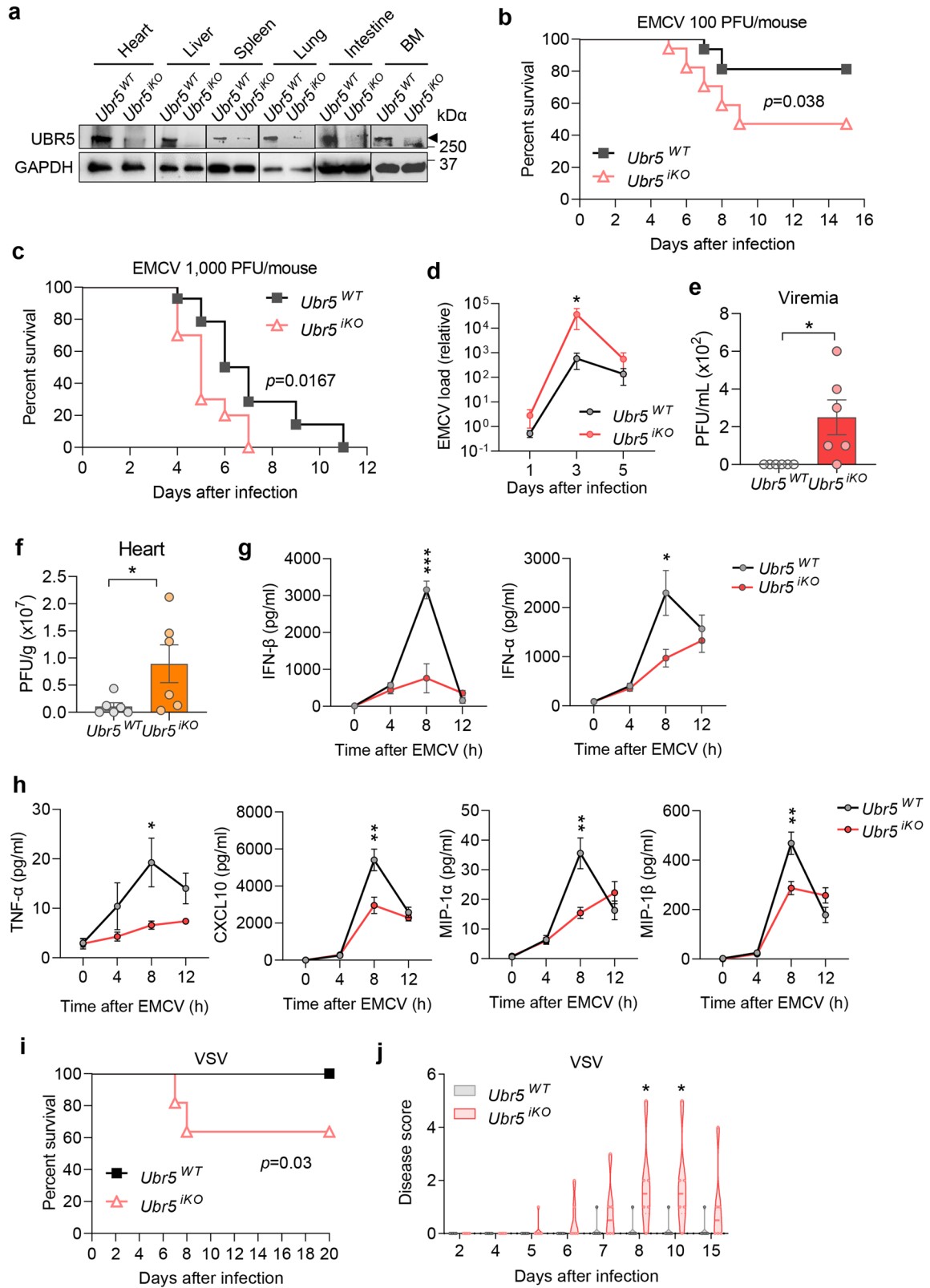

cells (Extended Data Fig. 7b), suggesting that UBR5 is dispensable for the MDA5 protein stability. Next, we examined the RLR transcript levels in the basal and poly (I:C)-stimulated states. The mRNA levels of *IFIH1* (gene symbol for MDA5) and *DDX58* (gene symbol for RIG-I) in the *UBR5*[−/−] were significantly lower than those in WT HEK293T and primary MEFs (Fig. 5g, h). These results suggested that UBR5 regulates RLR transcription.

## UBR5 promotes RLR transcription by inhibiting TRIM28

To pinpoint the mechanism of UBR5 action on RLRs, we searched for UBR5-interacting proteins. Of note, in two independent screenings for protein-protein interactions, UBR5 was shown to interact with TRIM28[23,24], which is an epigenetic repressor of *RLRs*[25]. Indeed, after revisiting the initial screening results with poly (I:C), we found that TRIM28 was one of the negative regulators of MDA5 signaling

**Fig. 4 | UBR5 is crucial for the induction of innate immune response and control of RNA virus pathogenesis in mice. a** The immunoblots of UBR5 in various tissues of age- and sex-matched littermates. Black triangle indicates the right band size. The survival curves of littermates infected with (**b**) 100 or (**c**) 1,000 plaque forming units (PFU) of EMCV intraperitoneally. In **b**, $n = 16$ for $Ubr5^{WT}$ and 17 for $Ubr5^{iKO}$, $p = 0.038$ (Log-Rank test); in **c**, $n = 14$ for $Ubr5^{WT}$ and 10 for $Ubr5^{iKO}$, $p = 0.030$ (Log-Rank test). Quantification of (**d**) EMCV RNA in the whole blood cells by qRT-PCR, **e** viremia and (**f**) viral loads in hearts by a plaque forming assay, in the mice infected with 100 PFU of EMCV. In **d**–**f**, $n = 6$ mice, mean ± S.E.M., two-tailed, unpaired non–parametric Mann–Whitney U test; *$p = 0.0411$ (**d**), *$p = 0.0152$ (**e**), *$p = 0.0411$ (**f**).

**g, h** Quantification of the serum type I IFN and cytokine concentrations by ELISA in the mice infected with $1 \times 10^7$ PFU of EMCV intravenously. $n = 7$ mice/group, mean ± S.E.M., two-tailed Student's $t$ test; ***$p = 0.0009$ for IFN-β, *$p = 0.0221$ for IFN-α (**g**); *$p = 0.0259$ for TNF-α, **$p = 0.0055$ for CXCL10, **$p = 0.0034$ for MIP–1α, **$p = 0.0049$ for MIP–1β (**h**). **i** The survival curves of age- and sex-matched littermates infected with $1 \times 10^7$ PFU of VSV intravenously. $n = 11$ mice/group, $p = 0.03$ (Log-Rank test). **j** The arbitrary morbidity score of VSV-infected animals. $n = 6$ mice/group, mean ± S.E.M., two-tailed Student's $t$ test; *$p = 0.0438$, *$p = 0.0438$ in sequence. Adjusted $p$ values are presented. All the mice used in this study were 8 weeks old. Source data are provided as a Source Data file.

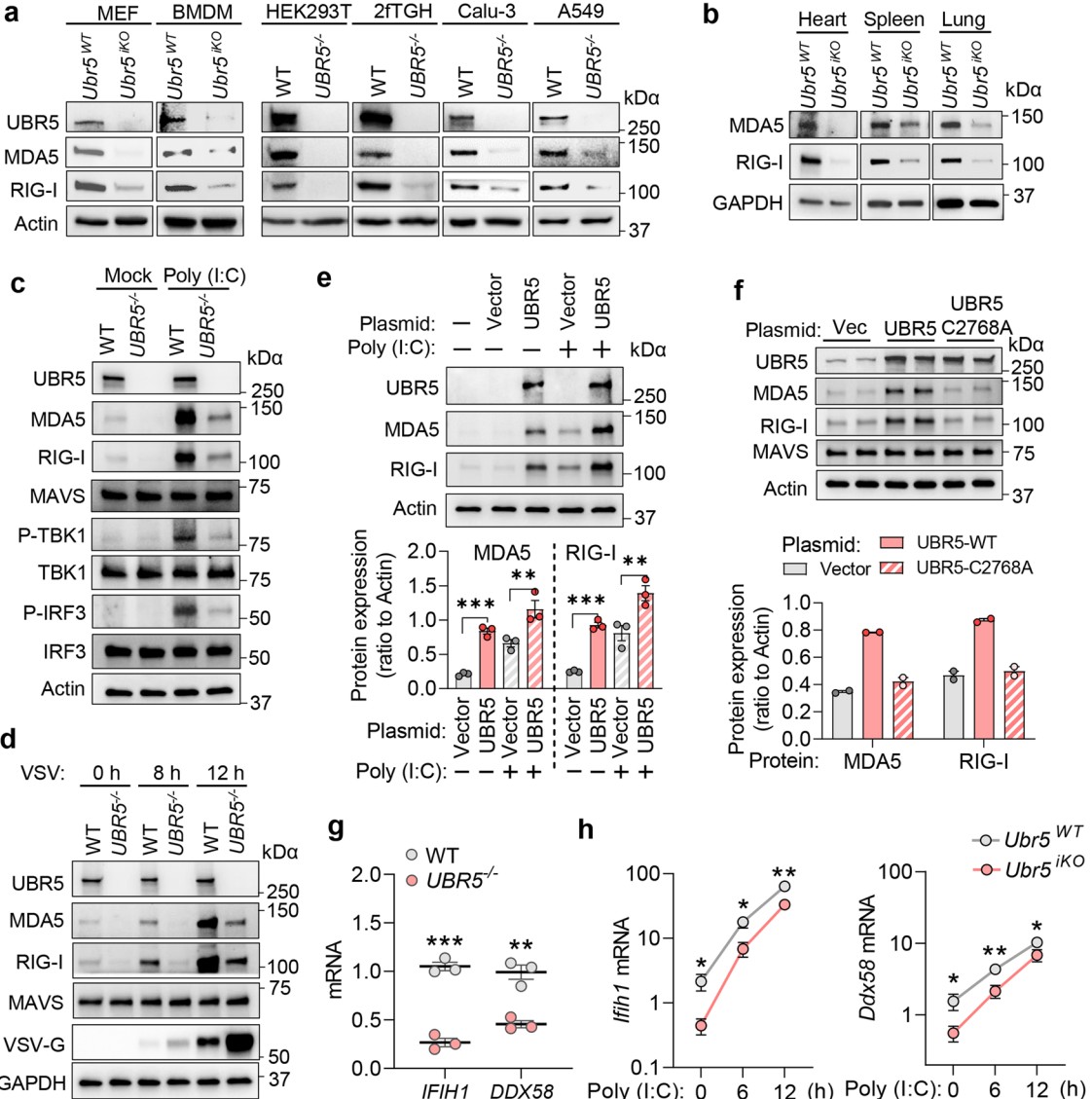

**Fig. 5 | UBR5 promotes RLR transcription.** The immunoblots of indicated proteins in (**a**) mouse primary MEFs, macrophages and various human cell lines, (**b**) the tissues of age- and sex-matched littermates. The immunoblots of indicated proteins in HEK293T cells (**c**) transfected with poly (I:C)/without (Mock) for 12 h, and (**d**) infected with VSV at a multiplicity of infection (MOI) of 0.5. **e** The immunoblots of indicated proteins in $UBR5^{-/-}$ HEK293T cells transfected with a UBR5 expression or vector plasmid for 24 h and then poly (I:C) (+) for 12 h. The bar chart indicates the ratios of MDA5/RIG-I band density to Actin. **f** The immunoblots of indicated proteins in WT HEK293T cells transfected with a vector, wild-type UBR5, or catalytic mutant C2768A plasmid. The bar chart indicates the ratios of MDA5/RIG-I band density to Actin. $n = 2$ biologically independent experiments. qRT-PCR

quantification of the *IFIH1/DDX58* (gene symbol for MDA5/RIG-I) mRNA levels in (**g**) HEK293T cells, and (**h**) MEFs transfected with poly (I:C). The data are representative of three independent experiments (**a**, **c**, **d**) or tissues from two mice (**b**) with similar results. The data shown in (**e**) are from one representative experiment of $n = 3$ biological independent experiments, mean ± S.E.M, ordinary one-way ANOVA with Dunnett's test, ***$p = 0.0008$, **$p = 0.0036$ for MDA5; ***$p = 0.0009$, **$p = 0.0023$ for RIG-I. Multiplicity adjusted $p$ values are presented. Data shown in **g**, **h**: mean ± S.E.M, two-tailed Student's $t$ test, $n = 3$ biologically independent experiments; ***$p = 0.0004$, **$p = 0.0028$ in **g**; for **h**: *$p = 0.0493$, *$p = 0.0458$, **$p = 0.0017$ in sequence for *Ifih1*; *$p = 0.0286$, **$p = 0.0086$, *$p = 0.0472$ in sequence for *Ddx58*. Adjusted $p$ values are presented. Source data are provided as a Source Data file.

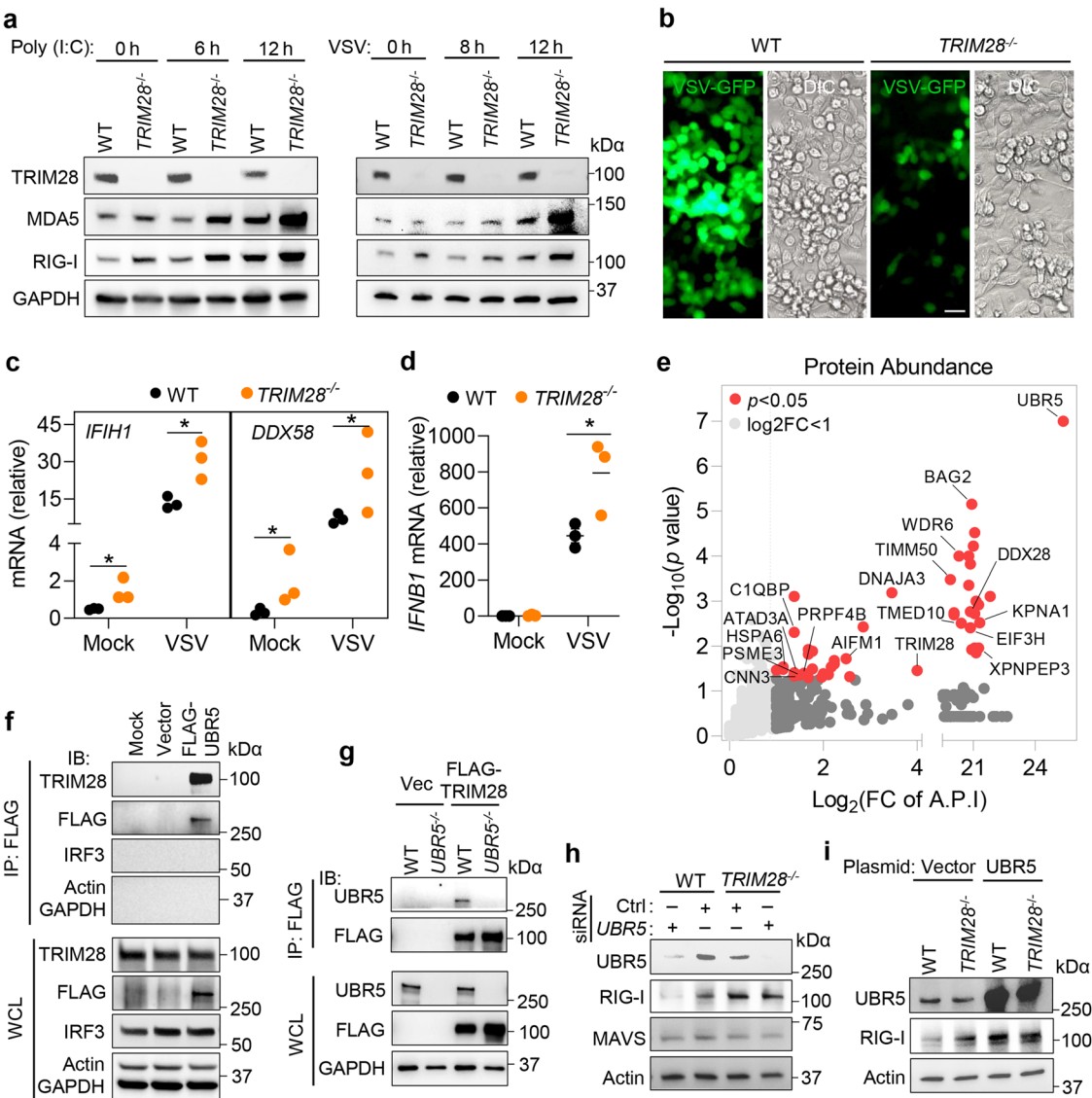

**Fig. 6 | UBR5 interacts with TRIM28, an epigenetic repressor of RLR. a** The immunoblots of indicated proteins in 2fTGH cells transfected with poly (I:C) or infected with VSV-GFP at a MOI of 0.5. **b** The fluorescent images of VSV-GFP in 2fTGH cells at 12 h *p.i.* Scale bar: 50 μM. Quantification of the (**c**) *IFIH1/DDX58* (gene symbol for MDA5/RIG-I) and (**d**) *IFNB1* mRNA levels by qRT-PCR in 2fTGH cells infected with VSV for 12 h. Data shown in **c** and **d** are presented as mean ± S.E.M, two-tailed Student's *t* test, *n* = 3 biologically independent experiments; *\*p = 0.0492, \*p = 0.0189* in sequence for *IFIH1*, *\*p = 0.0286, \*p = 0.0483* in sequence for *DDX58* in **c**; *\*p = 0.0489* in **d**. Adjusted *p* values are presented. **e** Scatter plot showing the 49 proteins identified by FLAG-UBR5- immunoprecipitated (IP)-mass spectrometer (MS) analysis. The 17 proteins validated at Harmonizome database are labeled. Positive proteins:

log$_2$(FC of A.P.I) > 1, two-tailed Student's *t* test with Benjamini−Hochberg, *p* < 0.05, FC of A.P.I: fold change of Average Precursor Intensity. **f**, **g** FLAG-UBR5 co-immunoprecipitated (IP) with endogenous TRIM28, and vice versa. FLAG-UBR5/TRIM28 or vector plasmid was expressed in HEK293T cells, and immunoprecipitated with an anti-FLAG antibody. The indicated proteins were immunoblotted (IB) with specific antibodies. WCL: whole cell lysate. The immunoblots of indicated proteins in HEK293T cells transfected with (**h**) a negative siRNA (Ctrl) or *UBR5* siRNA for 48 h, (**i**) with a UBR5 expression or vector plasmid for 24 h. The data are representative of two independent experiments with similar results (**a**, **f**–**i**). Source data are provided as a Source Data file.

(Fig. 1a, Supplementary Data 2). Both the constitutive and poly (I:C) or VSV-induced RLR protein levels were higher (Fig. 6a), while the VSV load was significantly repressed (Fig. 6b) in *TRIM28⁻/⁻* 2fTGH cells. Of note, the significant upregulation of RLRs by poly (I:C) in WT cells was observed only at 12 h (vs 0 h), but it was significant as early as 6 h (vs 0 h) in *TRIM28⁻/⁻* cells (Fig. 6a). Moreover, the RLR mRNA levels were higher before and after VSV infection (Fig. 6c), which led to enhanced *IFNB1* mRNA expression in *TRIM28⁻/⁻* cells after VSV challenge (Fig. 6d).

Next, we performed a Ultra performance liquid chromatography-tandem mass spectrometer (UPLC-MS/MS) to confirm the interaction of UBR5 and TRIM28 while also identifying new interactors of UBR5,

including 17 being reported in the Harmonizome protein-protein interaction database (Fig. 6e, Supplementary Data 5)[26]. These potential target proteins were related to transcription activity, ubiquitin-like protein ligase binding and viral infection pathways (Extended Data Fig. 8a, b). We then further validated the association of UBR5 and TRIM28 by co-immunoprecipitating the endogenous TRIM28 with FLAG-UBR5. This interaction was specific because IRF3/ACTIN/GAPDH were not pulled down (Fig. 6f). Conversely, FLAG-TRIM28 was able to pull down endogenous UBR5 (Fig. 6g). Next, we investigated the functional interaction between UBR5 and TRIM28. To this end, we employed an siRNA to silence *UBR5* in both the WT and *TRIM28⁻/⁻* HEK293T cells. The negative control (Ctrl) siRNA was a universal

non-targeting one. Consistent with the result from *UBR5*[−/−] cells, silencing UBR5 with the *UBR5* siRNA led to a reduction in the RIG-I protein level in WT cells. However, the *UBR5* siRNA showed no effect on the RIG-I level in the *TRIM28*[−/−] cells (Fig. 6h). Furthermore, overexpression of UBR5 enhanced RIG-I expression in the WT cells, but not in the *TRIM28*[−/−] cells (Fig. 6i). The above results suggested that UBR5 regulates RLR transcription via TRIM28.

## UBR5 inhibits auto-SUMOylation of TRIM28 by K63-linked ubiquitination

Like UBR5, TRIM28 is a large nuclear protein (Extended Data Fig. 8c) with multi-domains. TRIM28 is recruited to specific DNA sequences by KRAB-ZNF (Kruppel Associated Box-Zinc Finger protein) transcription factors, where it undergoes intramolecular SUMOylation (small ubiquitin-like modifier)[11]. SUMOylation is required for TRIM28 to recruit histone modifiers to form the H3K9me3 mark on nearby nucleosomes together with deacetylation of histone proteins and chromatin compaction[27]. Importantly, ubiquitination inhibits the SUMOylation of newly synthesized, mostly chromatin-binding nuclear proteins[28]. Therefore, we postulated that UBR5 mediates ubiquitination of TRIM28 that prevents TRIM28 from auto-SUMOylating and inhibiting the RLR promoters during viral infection. We first examined if UBR5 mediates TRIM28 protein degradation and vice versa. The TRIM28 protein level was normal in the *UBR5*[−/−] cells, as was UBR5 in *TRIM28*[−/−], compared to that in WT cells (Extended Data Fig. 8d), suggesting that UBR5 is dispensable for the TRIM28 protein stability. Next, we investigated if TRIM28 SUMOylation was affected in *UBR5*[−/−]. Indeed, we observed SUMOylated TRIM28 (HA/Myc blots in the IP) in WT cells, which modestly decreased after poly (I:C) treatment, suggesting depression of the TRIM28-imposed brake on RLR transcription (Fig. 7a). A similar trend was also noted in *UBR5*[−/−] cells. Regardless of poly (I:C) treatment, TRIM28 SUMOylation was consistently more in the *UBR5*[−/−] than that in WT cells. This phenomenon was valid for all three major SUMOs (Fig. 7a). We then examined TRIM28 polyubiquitination and its linkage using FLAG-TRIM28 and HA-WTUb, K63Ub or K48Ub only (all the other K residues are mutated to R, Extended Data Fig. 9a). Both the total and K63-linked, but not K48-linked polyubiquitination of FLAG-TRIM28 was largely reduced in *UBR5*[−/−], compared to that in WT cells (Fig. 7b). Reconstitution of UBR5 expression in *UBR5*[−/−] cells restored K63-linked ubiquitination of TRIM28 (Extended Data Fig. 9b). We next evaluated the UBR5-TRIM28 interaction, TRIM28 ubiquitination and SUMOylation during poly (I:C) treatment with FLAG-TRIM28. Notably, the amount of endogenous UBR5 immunoprecipitated by FLAG-TRIM28 was increased after the poly (I:C) treatment, as was the endogenous K63-linked ubiquitination of TRIM28 in WT cells (Fig. 7c). Deletion of UBR5 then lowered the K63-linked ubiquitination of TRIM28 before and after the poly (I:C) treatment. We also detected no difference in K48-linked ubiquitination of TRIM28 between the *UBR5*[−/−] and WT cells. Notably, the SUMOylated TRIM28 level was higher in *UBR5*[−/−] than that in WT cells before and after the poly (I:C) treatment (Fig. 7c). Lastly, we examined the endogenous TRIM28-UBR5 interaction, SUMOylation and ubiquitination of TRIM28 during VSV infection by immunoprecipitating endogenous TRIM28. In WT cells, the amount of UBR5 immunoprecipitated by TRIM28 increased significantly after VSV infection, which coincided with upregulated K63-linked ubiquitination, while reduced SUMOylation of TRIM28. The level of K63-ubiquitinated TRIM28 was always lower, while SUMOylated TRIM28 was consistently higher, in *UBR5*[−/−] than that in WT cells (Fig. 7d, e). To identify the UBR5-mediated ubiquitination site within TRIM28, we performed FLAG-TRIM28 immunoprecipitation from WT and *UBR5*[−/−] cells with/or without poly (I:C) for 6 h and identified ubiquitin modifications by UPLC-MS/MS. Of all the ubiquitinated lysine residues of TRIM28, only the rate of K507 ubiquitination increased following poly (I:C) in WT cells, but remained lower in *UBR5*[−/−] before and after poly (I:C), suggesting K507 is the

UBR5 target (Fig. 7f; Supplementary Data 6; Extended Data Fig. 10). To validate this, we next generated K507R mutant TRIM28. Since K779 in the bromodomain is the predominant SUMOylation site, we included K779R as a control[27] (Fig. 7g). Mutation of K507 significantly reduced UBR5-induced K63-linked ubiquitination of TRIM28, while K779R mutation failed to do so (Fig. 7h). As a negative control, ligase deficient UBR5 C2768A mutant failed to ubiquitinate either WT or mutant TRIM28. Compared to the C2768A mutant, WT UBR5 repressed the SUMOylation of WT, but not K507R mutant TRIM28. As expected, the TRIM28 K779R control was resistant to SUMOylation (Fig. 7h). These data strongly suggest that UBR5-mediated ubiquitination of K507 in TRIM28 blocks its auto-SUMOylation.

TRIM28 binds the RLR promoter DNA to impose chromatin compaction and inhibit RLR transcription[25]. We then asked if these modifications influence TRIM28 binding to RLR promoter DNA. We carried out a chromatin immunoprecipitation (ChIP) assay with an anti-TRIM28 antibody and noted that the amount of the RLR promoter DNA bound by TRIM28 was reduced after VSV infection in both WT and *UBR5*[−/−] respectively, but it was always higher in *UBR5*[−/−] than that in WT cells before (Mock) and after VSV infection (Fig. 7i). To confirm this in an unbiased manner, we sequenced the TRIM28-bound DNA and observed that ~5.8% of TRIM28-bound sites were enriched in promoter regions in both WT and *UBR5*[−/−] cells following VSV infection (Fig. 7j). Approximately 1933 gene promoters were enriched by over 10-fold enrichment (relative to input whole genome) in *UBR5*[−/−] cells, and over 2-fold relative to WT cells (Supplementary Data 8). Of note, TRIM28-bound *IFIH1/DDX58* promoter DNA was much more in *UBR5*[−/−] than that in WT cells. Intriguingly, we noted a similar trend for several related genes of IFN-I signaling, including *ISG20*, *CXCL10*, *MX1* and *IFIT1* (Fig. 7k), consistent with a recent publication[25]. However, we did not observe other known viral PRRs (*MB21D1*, *TLR3/7/9*, *NLRP*) or major components of the RLR pathways (*MAVS*, *TBK1*, *IRF*, *IFN* etc.) were transcriptionally regulated by TRIM28 (Supplementary Data 8). These results suggested that although UBR5 could potentially regulate many genes via TRIM28 during RNA virus infection, it preferably targets RLRs and several ISGs in the context of PRR-IFN signaling.

## The RLR-IFN axis is one of the primary common targets of UBR5 and TRIM28

Both UBR5 and TRIM28 are profoundly involved in transcriptional control[10,11], thus, the UBR5-TRIM28 axis could regulate genes critical for PRR-IFN responses, in addition to RLRs. To address this, we performed RNA-seq analyses of WT, *UBR5*[−/−] and *TRIM28*[−/−] HEK293T cells treated with/or without poly (I:C). In untreated *UBR5*[−/−] cells, 553 genes were significantly downregulated [Log2 fold change (FC) < −1] including *IFIH1* (Log2FC: −1.74, $p < 0.05$); *DDX58* was modestly downregulated (Log2FC: −0.62, $p < 0.05$) compared to WT (Fig. 8a; Supplementary Data 9). These results were largely consistent with the PCR data (Fig.5g). The downregulated genes were enriched in RNA Polymerase III Chain Elongation (Supplementary Data 9). After poly (I:C) stimulation, downregulated genes (*UBR5*[−/−] vs WT) (Log2FC < −1, $p < 0.05$) were enriched primarily in IFN-I signaling and TRAF3-dependent IRF activation pathways including *IFNB1*, *DDX58* and *IFIH1* etc. (Fig. 8b–d). To strengthen that UBR5 regulates RLR expression via TRIM28, we next analyzed the overlapping differentially expressed genes (DEGs) between *UBR5* and *TRIM28* deficient cells stimulated with poly (I:C). Considering the opposite role of UBR5 and TRIM28 in RLR expression, we compared the downregulated genes from *UBR5*[−/−] with the upregulated genes from *TRIM28*[−/−] cells. There were 42 overlapping DEGs, 9 of which were ISGs including *DDX58*, and 23 genes primarily involved in RNA Polymerase II transcription including 22 Zinc Finger (ZNF) transcripts and BATF2 (Fig. 8e). The DEGs significantly upregulated in *TRIM28*[−/−] vs WT cells were enriched predominantly in IFN-I signaling and RNA polymerase II transcription (Fig. 8f; Supplementary Data 9).

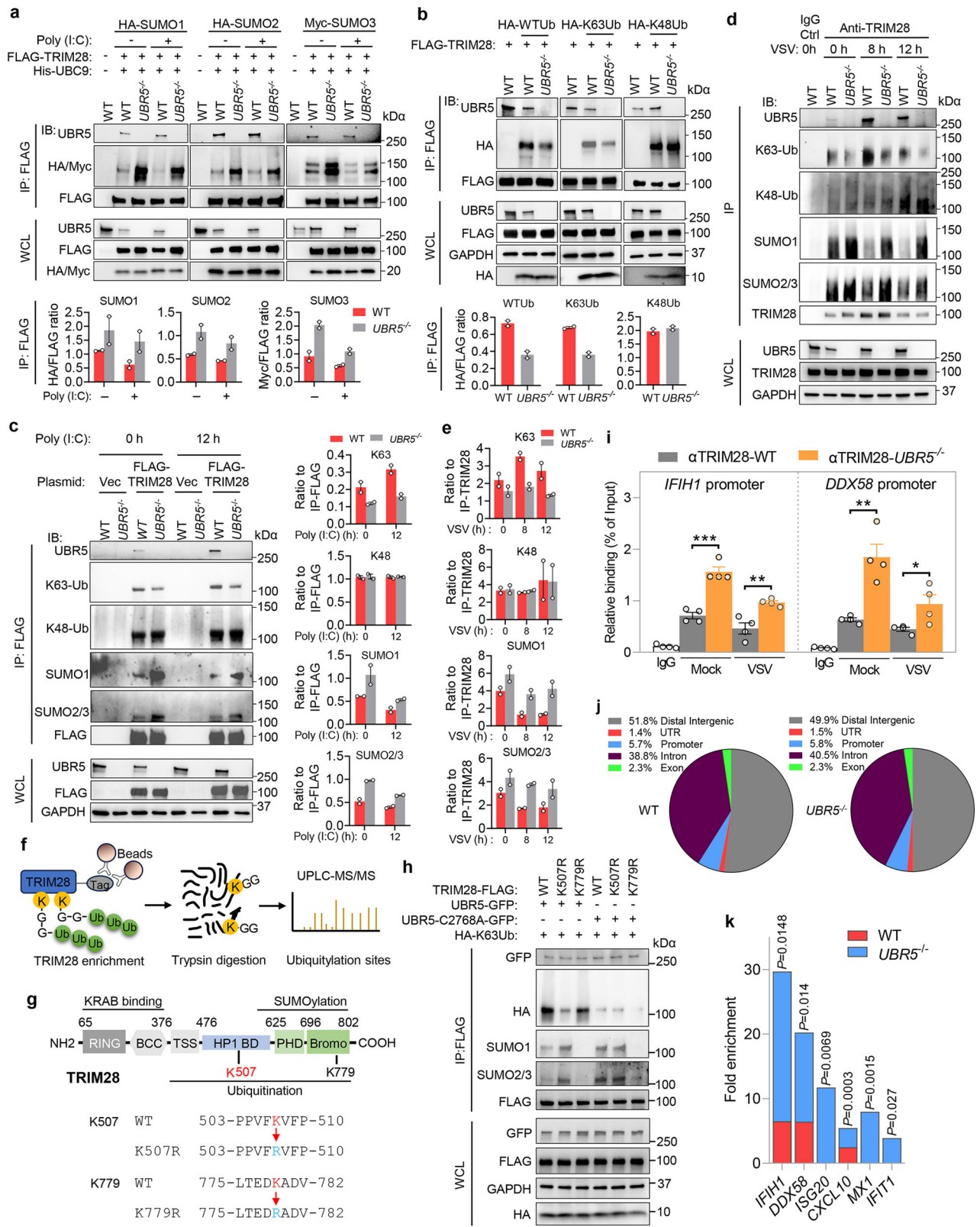

The IFNAR1-JAK-STAT1/2 pathway is critical for the upregulation of RLR expression during viral infection. To see if UBR5-TRIM28 regulates this pathway, we performed RNA-seq analyses of WT, *UBR5⁻/⁻* and *TRIM28⁻/⁻* HEK293T cells treated with recombinant human IFN-β for 6 h. We observed no differences in RLR or other conventional ISG expression[29] between WT and *UBR5⁻/⁻* cells (Extended Data Fig. 11a, b; Supplementary Data 10), consistent with the primary CRISPR

screening results (Fig. 1d) showing that IFN-β-induced ISRE-Luc was no different between WT and *UBR5⁻/⁻* cells. These results suggested that UBR5 is dispensable for the IFNAR-JAK-STAT1/2 pathway. Notably, in *TRIM28⁻/⁻* cells, a small number (22) of conventional ISGs including *DDX58*, *MX1/2*, *OAS1/2/3*, and *IFIT2/3* were upregulated, suggesting that TRIM28 selectively targets individual ISGs rather than the entire IFNAR-JAK-STAT1/2 pathway (Supplementary Data 11). Overall, these

**Fig. 7 | UBR5 promotes K63-linked ubiquitination but inhibits SUMOylation of TRIM28. a** WT and *UBR5⁻/⁻* HEK293T cells were transfected with FLAG-TRIM28, His-UBC9 and HA- or Myc -SUMO plasmids for 24 h, then without (Mock) or with poly (I:C) for 12 h. FLAG-TRIM28 was immunoprecipitated (IP) with an anti-FLAG antibody, and the IP and whole cell lysate (WCL) were immuno-blotted (IB) for the indicated proteins with specific antibodies. **b** WT and *UBR5⁻/⁻* HEK293T cells were transfected with FLAG-TRIM28, HA-tagged WT, K48 or K63-only ubiquitin (Ub) for 24 h. The IP and IB was performed as above. **c** WT and *UBR5⁻/⁻* HEK293T cells were transfected with FLAG-TRIM28 or vector for 24 h, then with poly (I:C) for 12 h. The IP and IB was carried out as above. The bar chart in **a**–**c** indicates the ratios of the indicated protein band density. *n* = 2 biologically independent experiments. **d** WT and *UBR5⁻/⁻* HEK293T cells were infected with VSV at a MOI of 0.5. Endogenous TRIM28 was immunoprecipitated with an anti-TRIM28 antibody. **e** The bar chart indicates the ratios of the indicated protein band density in **d**. *n* = 2 biologically independent experiments. **f** The method to identify ubiquitinated sites within

TRIM28. **g** The method for generating K507R and K779R mutants of TRIM28. **h** HEK293T cells were transfected with FLAG-TRIM28 (WT, K507R, K779R), GFP-UBR5, GFP-UBR5-C2758A mutant and HA-K63Ub plasmids for 24 h, then FLAG-TRIM28 was immunoprecipitated with an anti-FLAG antibody, and the IP and WCL were immunoblotted for the indicated proteins with specific antibodies. **i**–**k** WT and *UBR5⁻/⁻* HEK293T cells were infected with VSV at a MOI of 0.5 for 8 h. Chromatin immunoprecipitation (ChIP) was performed with an anti-TRIM28 antibody. **i** The TRIM28-bound RLR promoter DNA was quantified by qPCR and normalized to its input. Bar: mean ± S.E.M, two-tailed Student's *t* test, *n* = 4 biologically independent samples, ***p = 0.0006, **p = 0.0094 for *IFIH1*; **p = 0.0029; *p = 0.0382. Adjusted *p* values are presented. **j** ChIP-seq analysis and genomic annotation of TRIM28-bound sites in infected cells. UTR: untranslated regions. **k** Fold enrichment in TRIM28-bound promoter regions of select ISGs. The *P*-value was generated in Peak calling statistics using a Poisson distribution with local lambda estimate. Source data are provided as a Source Data file.

results indicated that TRIM28 targets a broader spectrum of genes than UBR5 does and UBR5 is dispensable for RLR induction by IFNAR-JAK-STAT1/2.

## Discussion

The UBR5 protein is highly conserved in metazoans (murine and human proteins are 98% identical), may mediate K48-liked poly-ubiquitination and degradation of many proteins that function in the DNA damage response, metabolism, transcription, and apoptosis[10]. UBR5 may also mediate K63-linked polyubiquitination and activation of the NF-κB cofactor AKIRIN2[30]. In the context of viral infection, UBR5 could regulate viral protein turnover or could be hijacked by viruses to interfere with cellular processes. Accordingly, UBR5 suppresses expression of papillomavirus type 8 E6/E6AP ubiquitin E3 ligase complex[31], degrades human T cell leukemia virus (HLTV-1) HBZ protein[32], and mediates K29-ubiquitination of hepatitis B virus HBc protein[33]. On the other hand, UBR5 is hijacked by human immunodeficiency virus 1 (HIV-1) Vpr protein to inhibit telomerase activity[34] or disrupt centrosome homeostasis[35]. Herein, we have identified and characterized a role of UBR5 in the major antiviral immune signaling pathways, RLRs. We have demonstrated that UBR5 is essential for RLR signaling and control of RNA virus infection ex vivo and in vivo (Figs. 2–4). Mechanistically, UBR5 promotes RLR transcription by depressing the TRIM28-imposed brake on the RLR promoter (Figs. 6 and 7). In favor of this concept, both proteins are primarily localized to the nucleoplasm, and are established transcriptional regulators[10,11]. In particular, TRIM28 is known to bind the RLR promoter regions and repress their transcription epigenetically[25]. Moreover, TRIM28 seems to indirectly regulate RLR signaling. For example, TRIM28 deficiency can unleash expression of endogenous retroviral elements (ERVs)[36], which serve as a dsRNA source to activate RLR signaling[37]. Although expression of a few ISGs (including RIG-I) were upregulated, type I IFNs remained largely unchanged in TRIM28-deficient cells[37], suggesting that the unleashed ERVs alone are unable to induce a significant type I IFN response and that TRIM28 may directly regulate the transcription of these ISGs. Indeed, TRIM28 does selectively target some conventional ISGs, but not the whole IFNAR-JAK-STAT1/2 pathway (Supplementary Data 11). Nonetheless, both mechanisms of TRIM28 action, *i.e.*, ERV and RLR transcription, may operate together and form a positive feedback loop during RNA virus infection.

Both UBR5 and TRIM28 are profoundly involved in transcriptional control[10,11]. Indeed, our whole transcriptome analyses have shown that UBR5/TRIM28 upregulates/downregulates hundreds of genes, including overlapping genes in the RLR-IFN pathway and RNA Polymerase II transcription. In the context of PRR and IFN-I signaling, the UBR5-TRIM28 axis preferably targets the RLRs, largely consistent with a recent study showing that TRIM28 preferably targets the RLR promoters[25]. This selectivity of TRIM28 for RLRs during RNA virus infection is likely determined by UBR5, evidenced by the enhanced interaction between

UBR5 and TRIM28, K63-linked ubiquitination of TRIM28, while reduced SUMOylation of TRIM28 following VSV infection (Fig. 7d). However, TRIM28 may target some ISGs downstream of IFNAR-JAK-STAT1/2, but independently of UBR5. In addition to ubiquitination, TRIM28 is regulated by phosphorylation. Ataxia-Telangiectasia Mutated (ATM) kinase, a member of nuclear phosphatidylinositol-3 kinase–like (PIKK) family, mediates phosphorylation of TRIM28, resulting in chromatin relaxation[11]. Notably, type I IFNs drive ATM-dependent TRIM28 phosphorylation[38]. Thus, phosphorylation may be the dominant regulation of TRIM28 when only JAK-STAT1/2 pathway is activated. In agreement with this, UBR5 is dispensable for RIG-I expression induced by TBK1 overexpression (Extended Data Fig. 6b), which activates IRF3/7, type I IFN expression and subsequent JAK-STAT1/2 pathway. Intriguingly, UBR5 is crucial for RIG-I expression induced by RIG-I/MAVS overexpression, which leads to more complex signaling events than just TBK1, for example, NF-κB and MAPK etc. Therefore, our results imply that UBR5 activity requires upregulation by RLR-MAVS signaling, further disengaging TRIM28 imposed-brake on RLR transcription.

TRIM28 was shown to epigenetically inhibit IRF5 function and IRF5-mediated inflammatory gene expression in macrophages[39]. However, the expression of IRF5-targeted genes (*IL6, IL12, IL23, TNF*) was normal in TRIM28-deficient HEK293T cells, likely due to low expression of IRF5. Although IRF5 is dispensable for type I IFN induction, it could influence viral pathogenesis in vivo via inducing inflammatory responses[40]. Further defining the role of UBR5 in IRF5 function in macrophages will be an interesting topic to purse in the future.

UBR5 plays a moderate role in the control of HSV-1 replication too, but largely independently of the viral DNA-sensing pathways and type I IFN response (Fig. 3h, i). This is not surprising because UBR5-TRIM28 could also directly regulate the transcription of a few *ISG*s, of which, ISG20, MX1, and IFIT1 can directly interfere with the replication of many viruses[41–43] including HSV-1[44]. Additionally, UBR5 could suppress viral gene expression[31], degrade viral proteins via ubiquitination[32], or alter viral protein functions via K29-linked ubiquitination[33]. However, these UBR5 features are likely virus specific. Therefore, UBR5-TRIM28 may control RNA virus infection preferably by regulating RLR-IFN and by directly regulating the expression of a few antiviral effector ISGs.

UBR5 has been well known to mediate K48-linked polyubiquitination of transcriptional regulators[10]. Herein, we have clearly shown that UBR5 promotes K63-linked ubiquitination of TRIM28, leading to inhibition of its intramolecular SUMOylation and depression of RLR transcription following virus infection (Fig. 7). Like ubiquitination, SUMOylation specifically targets lysine residues, thus the two processes may compete for the same lysine residue and differentially altering the target protein function[45]. Alternatively, non-competing ubiquitination of a protein may cause a conformational change that impacts its SUMOylation and functionality. In this study, we have identified K507 of TRIM28 is the UBR5-mediated ubiquitination site (Fig. 7f–h). Though not the key SUMOylation site[27], K507 lies in the

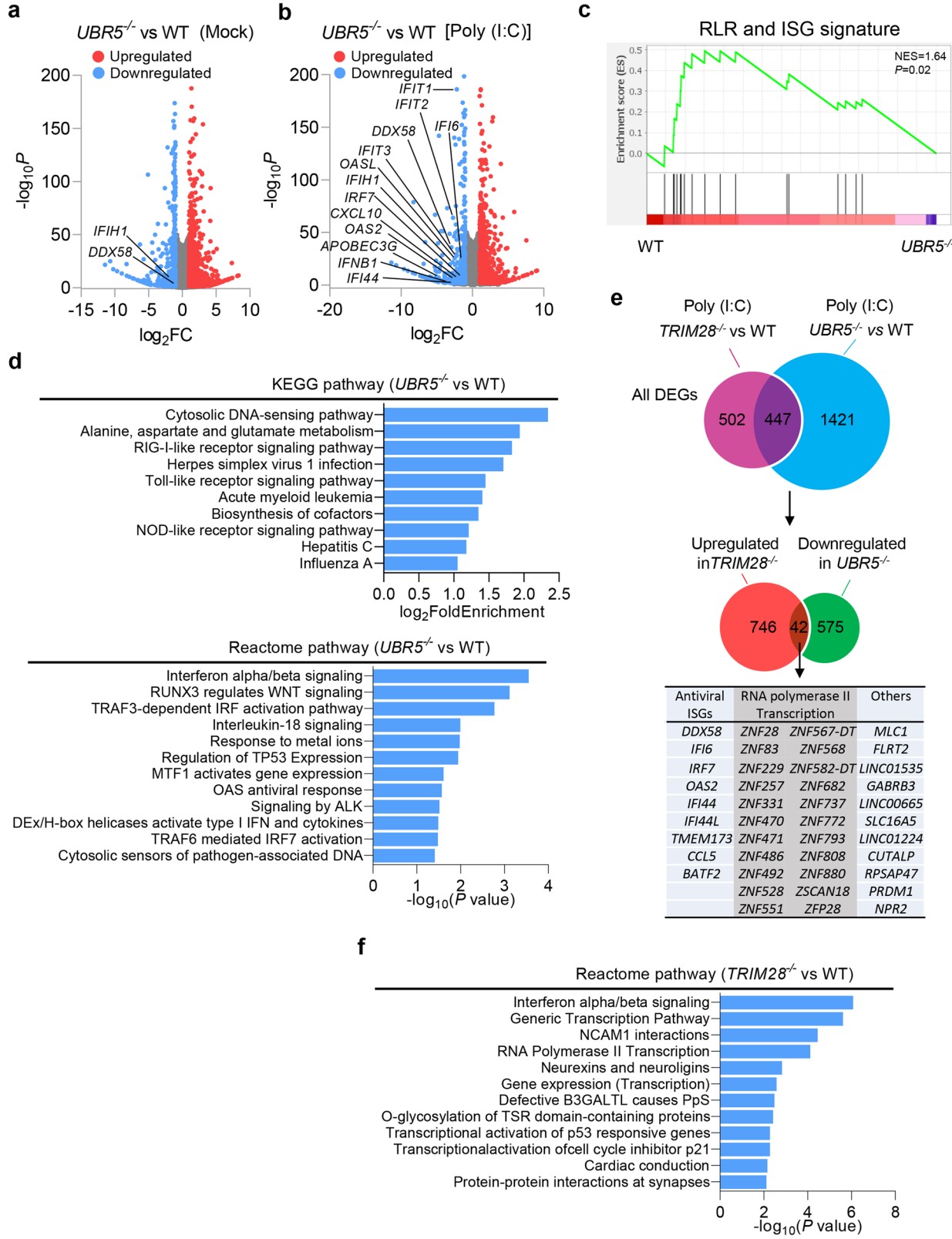

**Fig. 8 | The RLR-IFN axis is one of the primary common targets of UBR5 and TRIM28.** **a**, **b** Volcano plots of differentially expressed genes (DEGs) in *UBR5*$^{-/-}$ versus WT cells with/or without poly (I:C)-stimulation. Red and blue represent significant DEGs of upregulated (log$_2$FC ≥ 1, $p < 0.05$) and downregulated (log$_2$FC ≤ −1, $p < 0.05$), respectively. DESeq2 was used to perform a comparison of gene expression between defined groups, the Wald test was used to generate log$_2$FC and $p$-values adjusted with the Benjamini–Hochberg. **c** GSEA plot of a significant gene set associated with RLR pathway and ISGs, $p$ values were calculated by one-way ANOVA with Tukey's post hoc comparison. **d** Top KEGG and Reactome pathways enriched from downregulated DEGs in *UBR5*$^{-/-}$ versus WT cells upon poly (I:C)-stimulation ($p < 0.05$, right-tailed Fisher's exact $t$ test with Benjamini & Hochberg). **e** Venn diagram revealing 42 genes overlapping between the DEGs upregulated in *TRIM28*$^{-/-}$ and downregulated in *UBR5*$^{-/-}$ cells. **f** Top Reactome pathways enriched from upregulated DEGs in *TRIM28*$^{-/-}$ versus WT cells upon poly (I:C)-stimulation. Source data are provided as a Source Data file.

conserved heterochromatin protein 1 (HP1)-binding domain (HP1BD) corresponding to amino acid residue 483-510[46]. Of note, HP1BD is also critical for the nuclear targeting, repression function and SUMOylation of TRIM28[27]. Therefore, K507 could be a key amino acid residue for HP1BD functionality. Indeed, UBR5 inhibits not only TRIM28 SUMOylation but also binding to the RLR promoters (Fig. 7).

It seems paradoxical for UBR5 (positive) and TRIM28 (negative) to be in the same axis to regulate RLR transcription. Normally, the basal transcription of RLRs remains very low because RLRs may be activated by self RNA, leading to autoinflammation. In particular, the two CARDs of MDA5 are constitutively exposed[1], allowing for MAVS activation in the absence of ligands that could contribute to the pathogenesis of autoimmune diseases[3–5]. This would be further damaging if the constitutive MDA5 transcription level were high. However, during RNA virus infection, the RLR transcription is rapidly mobilized before the secondary type I IFN response e.g., JAK-STAT1/2 is activated. UBR5-TRIM28 may serve as a rheostat to 1) minimize steady-state RLR transcription by TRIM28, and 2) to depress the TRIM28 brake rapidly upon viral infection. Intriguingly, only a small proportion of a protein needs be SUMOylated to achieve its maximal effect, a phenomenon known as the "SUMO enigma"[47]. This feature could render the UBR5-mediated finetuning of TRIM28 ubiquitination and SUMOylation to achieve a large effect on RLR transcription.

In summary, we have shown the crucial role of UBR5 in control of RNA virus infection and RLR transcription. Mechanistically, UBR5 mediates the K63-linked ubiquitination of TRIM28, repressing its SUMOylation and activating RLR transcription. Future work is needed to elucidate if UBR5-mediated ubiquitination influences epigenetic modifications of RLR promoter DNA and TRIM28 interaction with DNA modifiers.

## Methods

The current study complies with all relevant ethical regulations and the study protocol approved by the Institutional Biosafety Committee (IBC) of the University of Connecticut (UConn) and the Institutional Animal Care and Use Committee at UConn Health adhering to federal and state laws.

### Reagents and antibodies

The anti-GAPDH (Cat# 60004-1-Ig, 1:2000), UBR5 (Cat# 22782-1-AP, 1:200 for IF) and GFP (Cat # 50430-2-AP, 1:1000) were purchased from Proteintech Group, Inc (Rosemont, IL, USA). The mouse anti-TRIM28 antibody (20A1, Cat # 619302, 1:1000) was purchased from BioLegend (San Diego, CA, United States). The anti-UBR5 antibody (Cat# A300-573A, 1:1000) was purchased from Bethyl Laboratories, Inc (Montgomery, TX, United States). MG132 (Cat# 2194) and the anti- Tubulin (Cat# 2144, 1:2000), Actin (Cat#4967, 1:2000), K63-linked polyubiquitin (D7A11, Cat# 5621,1:500), K48-linked polyubiquitin (D9D5, Cat# 12805, 1:1000), SUMO-1 (Cat# 4930, 1:500), SUMO -2/3 (18H8, Cat# 4971, 1:500), His-Tag (D3I1O, Cat# 12698, 1:1000), HA-Tag (C29F4, Cat# 3724, 1:1000), RIG-I (D14G6, Cat# 3743, 1:1000), MDA5 (D74E4, Cat# 5321, 1:1000), MAVS (D5A9E, Cat# 24930, 1:1000), UBR5 (D6O8Z, Cat# 65344, 1:1000), IRF3 (D6I4C, Cat # 11904, 1:1000) and STING (D2P2F, Cat#13647, 1:1000) were purchased from Cell Signaling Technology (Danvers, MA, United States). The mouse anti-Myc antibody (9E10, Cat# TA150121, 1:1000) and human UBR5 siRNA oligo duplex (Locus ID 51366, Cat# SR309739) were obtained from OriGene Technologies, Inc (Rockville, MD, United States). TransIT-X2 Dynamic Delivery System (Cat# MIR6005) was obtained from Mirus Bio LLC (Madison, WI, United States). IFN stimulatory DNA derived from Listeria monocytogenes genome (ISD, Cat# tlrl-isdn), 5′ triphosphate hairpin RNA (Cat# tlrl-hprna), and high molecule weight polyinosine-poly cytidylic acid [HMW poly (I:C),1.5−8 kb, Cat# tlrl-pic], Poly(U) (Cat# tlrl-sspu), FSL-1 (Cat# tlrl-fsl) and CpG DNA (Cat# tlrl-2395) were products of Invivogen (San Diego, CA, United States). The anti-FLAG

M2 magnetic beads (Clone M2, Cat# M8823), FLAG (Clone M2, Cat # F1804-200UG, 1:1000) and VSV-G antibody (Cat# V4888, 1:1000) and Lipopolysaccharides (O111:B4, Cat# L3024) were from Sigma-Aldrich (St. Louis, MO, United States). Anti-rabbit IgG, HRP-linked Antibody (Cat#7074, 1:5000), Anti-mouse IgG, HRP-linked Antibody (Cat#7076, 1:5000) were purchased from Cell Signaling Technology (Danvers, MA, United States). The anti-Myc magnetic beads (Clone 9E10, Cat# 88842), Donkey anti-Mouse Alexa Fluor™ Plus 488 (Cat# A32766, 1:200), Goat anti-rabbit Alexa Fluor™ Plus 594 (Cat# A32740, 1:200), DAPI Solution (Cat# 62248, 1:1000) and PureLink™ Genomic DNA Mini Kit (Cat# K182002) were from Thermo Fisher (Waltham, MA, United States). The recombinant human IFN-β (Cat#8499-IF) and human IFN-β DuoSet ELISA kit (Cat# DY814-05) were from R&D system, Inc (Minneapolis, MN, United States). The QuikChange II Site-Directed Mutagenesis Kit (Cat# 200523) was purchased from Agilent Technologies, Inc (Santa Clara, CA, United States). Alt-R™ Genome Editing Detection Kit (Cat# 1075932) and all of the primers used in this study were obtained from Integrated DNA Technologies, Inc. (Coralville, Iowa, United States). The Luciferase Assay (Cat# E1501) kit, Dual-Luciferase® Reporter Assay kit (Cat# E1910) and Reporter Lysis buffer (Cat# E397A) were purchased from Promega Corporation (Madison, WI, United States).

### Mouse model

All the animal procedures were approved by the Institutional Animal Care and Use Committee at UConn Health adhering to federal and state laws. Both sexes were used and included in analyses. $Ubr5^{flox/flox}$ was generated recently by Prof. Robert E. Hill[17] on a background of C57BL/6J and crossed with a tamoxifen inducible $ERT2$-Cre recombinase line (The Jackson Laboratory, Stock # 008463, background: C57BL/6J), to generate $ERT2$-Cre$^{+/-}$ $Ubr5^{flox/flox}$ mice. To induce global $Ubr5$ knockout in >6 weeks old mice, 1 mg of tamoxifen (dissolved in corn oil) was administered to each animal every other day, totaling five times. This line was designated inducible Ubr5 knockout ($Ubr5^{iKO}$) to distinguish it from constitutive knockouts. $ERT2$-Cre$^{+/-}$ $Ubr5^{flox/flox}$ mice treated with corn oil served as the wild-type control ($Ubr5^{WT}$). Tamoxifen was cleared for two weeks after the last dose. These mouse strains were housed in the specific pathogen-free animal facility at UConn Health. All mice were housed at an ambient temperature of approximately 24 °C, a humidity of 40-60%, and a light/dark cycle of 12 h. Genotyping was performed with genomic DNA and Choice Taq Blue Mastermix (Denville Scientific, Cat# CB4065-8) under the following PCR: 95 °C for 1 s, 34 cycles of 94 °C for 1 min, 60 °C for 30 s, 72 °C for 30 s, and then 72 °C for 7 min, 4 °C to stop. The genotyping primers were: wild type Ubr5 Forward 5′ GTTTCTGGCAAGGTT CAGTGC; Reverse 5′ CACACATGCTGCACAAACACATG; LoxP Ubr5 Forward 5′ CGCGAAGAGTTTGTCCTCAC; Reverse 5′ GCCTCGATC CTCCCTTTATC. The PCR reaction resulted in a product of 200 bp (WT) and/or 400 bp (LoxP). The Cre primers were common-5′ AAGG GAGCTGCAGTGGAGTA, WT reverse, 5′-CCGAAAATCTGTGGGAAGTC, and mutant reverse, 5′-CGGTTATTCAACTTGCACCA. The PCR reaction resulted in a product of 297 bp (WT) and/or 450 bp (Cre).

### Viral infection in mice

EMCV, VSV infection and disease score in mice was performed according to previous methods[16,48]. Virus stock was diluted properly in sterile phosphate-buffered saline (PBS). 8 weeks old age- and sex-matched littermates were infected with 100 plaque-forming units (PFU) or 1,000 PFU of EMCV intraperitoneally or $1 \times 10^7$ PFU intravenously. $1 \times 10^6$ PFU of VSV in 50 μL of PBS was injected into mice retro-orbitally. The animal morbidity and mortality were monitored twice a day for 15−20 days. The morbidity was recorded with a scale 0 to 5, 0 = no symptom, 1 = no plantar stepping in one hind leg, 2 = no plantar stepping in two hind legs or slight ankle movement in one hind leg, 3 = slight ankle movement in two hind legs or no ankle movement in

one hind leg, 4 = no ankle movement in one hind leg and slight in the other one and 5 = no ankle movement in both hind legs, almost loss of all movement ability. The mice that were scored for 5 were terminated for the humane endpoint.

## Plasmids

FLAG-TRIM28 (Item #124960)[49], pcDNA3 Myc-Sumo3 WT (Item #48964), pcDNA3 HA-Sumo1 WT (Item #48966), pcDNA3 HA-Sumo2 WT (Item #48967)[50], GFP-UBR5 ΔHECT (Item #52051)[51], pET28-His6-Ubc9 (Item #133909)[52], pCMV-Tag2B UBR5 (Item #37188), pCMV-Tag2B UBR5 C2768A (Item #37189), and pEGFP-C1-UBR5 (Item #37190)[22] were obtained from Addgene Inc. (Watertown, MA, United States). The plasmids encoding human MDA5, RIG-I, MAVS, TBK1, IRF3(5D), ISRE-firefly luciferase, actin promoter-driven renilla luciferase were previously described[53]. FLAG-TRIM28 K779R, K750R and K405R were made by site-directed mutagenesis. The HA-Ub, HA-K63Ub (all the other K are mutated to R) and HA-K48Ub plasmids were a kind gift of Dr. Rongtuan Lin[54].

## Cell culture, virus culture and titration

2fTGH is a human fibrosarcoma cell line (SKU 12021508, Sigma-Aldrich, St. Louis, MO 68178 United States). 2fTGH was transfected with an ISRE (Interferon Stimulated Response Element) -luciferase (firefly) reporter plasmid to generate a stable 2fTGH-ISRE-Luc line[53]. A549 (human lung carcinoma epithelial cell, Cat# CCL-185), Calu-3 (human lung adenocarcinoma epithelial cell, Cat# HTB-55), HEK293T cells (human embryonic kidney, Cat# CRL-3216), Vero cells (monkey kidney epithelial cells, Cat# CCL-81), and L929 cells (mouse fibroblast cells, Cat# CCL-1) were purchased from the American Type Culture Collection (ATCC) (Manassas, VA 20110, USA). A549, Calu-3, HEK293T/Vero and 2fTGH/L929 cells were grown in F-12K, Eagle's Minimum Essential Medium (EMEM), Dulbecco's modified Eagle's medium (DMEM) or Roswell Park Memorial Institute (RPMI) 1640, respectively supplemented with 10% fetal bovine serum (FBS) and antibiotics/antimycotics (ThermoFisher Scientific, Waltham, MA 02451, United States). These cell lines are not listed in the database of commonly misidentified cell lines maintained by ICLAC and have not been authenticated in our hands. However, they are routinely treated with MycoZAP (Lonza, Basel, Switzerland) and tested for mycoplasma contamination in our hands.

EMCV (Cat# VR-129B), HSV-1 (Cat# VR-1789) and VSV (Indiana strain, Cat# VR-1238) were purchased from American Type Culture Collection (ATCC) (Manassas, VA 20110, United States). Green fluorescence protein (GFP)-VSV was made by inserting a VSV-G/GFP fusion sequence between the VSV G and L genes[55]. These viruses were propagated in Vero cells and titrated by a plaque forming assay. Briefly, serially diluted (10-fold) viral samples were applied to confluent Vero cells in a 6-well plate at 37 °C for 2 h. The inoculum was then removed and replaced with 2 mL of complete DMEM medium with 1% SeaPlaque agarose (Lonza, Cat# 50100). After the medium solidified, the plate was incubated for 2-3 days at 37 °C, 5% $CO_2$. Plaques were visualized by Neutral Red staining (Sigma-Aldrich). Viral titers were expressed as plaque forming units (PFU)/mL or gram of tissues.

## Isolation and culture of primary cells

Bone marrow cells were differentiated into macrophages (BMDM) in L929-conditioned medium (RPMI 1640, 20% FBS, 30% L929 culture medium, 1x antibiotics/antimycotics) in a 10-cm Petri dish at 37 °C, 5% $CO_2$ for 5–7 days[16], with a change of medium every 2 days. Attached BMDMs were dislodged by pipetting and counted for plating in cell culture plates in plating medium (RPMI 1640, 10% FBS, 5% L929-conditioned medium, 1x antibiotics/antimycotics).

Mouse embryonic fibroblasts (MEFs) were obtained from pregnant ERT2-Cre+ Ubr5flox/flox mice of E12 to 14. Embryos were decapitated and eviscerated and then digested with trypsin for 15 min at 37 °C to release single cells. The cell suspension was filtered through a 70 μM strainer filter (Corning™ # 431751), cultured in RPMI 1640 medium supplemented with 10% FBS and antibiotics/antimycotics. One-half of the cells were treated with 2 μM of 4-hydroxyl (OH) tamoxifen for 4 to 5 days to generate the Ubr5iKO cells. The other half was treated with solvent (dimethyl sulfoxide, DMSO) and remained as Ubr5WT. After induction, the cells were further passaged three times and used for experimentation without tamoxifen.

## Generation of knockout cells by CRISPR-Cas9

Pre-designed, gene unique guide (g) RNA (Integrated DNA Technologies, Coralville, IA, United States) (Supplementary Data 1) was subcloned into a lentiCRISPR-v2 vector[13] and correct insertion was confirmed by sequencing. To generate lentiviral particles, each gRNA vector was transfected into HEK293T cells with the packaging plasmids pCMV-VSV-G[56] and psPAX2 (#12259, from the Didier Trono lab via Addgene, Watertown, MA 02472, United States). A half of the cell culture medium was replaced with fresh medium at 24 h and viral particles were collected at 48–72 h after transfection. The viral culture was cleared by brief centrifugation. Target cells (2fTGH-ISRE-Luc, A549, Calu-3 or HEK293T) at ~50% confluence were transduced with each lentiviral vector individually for 24 h and selected with 2 μg/mL of puromycin in 4-5 days. The sgRNAs for IFIH1, MAVS, STING, IFNAR1, TRIM28 and UBR5 in the Supplementary Data 3.

## T7 endonuclease I (T7EI) mismatch cleavage assay

The efficiency of genome editing by CRISPR-Cas9 was estimated with the Alt-R™ Genome Editing Detection Kit (IDT, Cat# 1075932). Briefly, genomic DNA was extracted using PureLink™ Genomic DNA Mini Kit (Thermo Fisher, Cat# K182002). The genomic DNA fragment containing a gRNA sequence of interest (~600 bp, different sizes upstream/downstream of the gRNA) was amplified by PCR with Choice Taq Blue Mastermix (Denville Scientific, Cat# CB4065-8). The PCR product was cleaved by T7EI exactly according to the product manual. The percentage of edited genome was calculated using GelAnalyzer 2010a software tool with formular: of %$gene editing$ =

$$(1 - \sqrt{1 - \frac{Band2 + Band3 + \ldots + BandX}{Band1 + Band2 + Band3 + \ldots + BandX}}) \times 100.$$ The PCR primers that amplify the CRISPR target site are in the Supplementary Data 4.

## E3 knockout library screening

Each batch of screening included WT and ~20 E3 knockout 2fTGH-ISRE-Luc lines. Cells in triplicate in a 48-well plate at 70% confluence were transfected with 0.4 μg/mL of ISD, 1 μg/mL of high molecular weight poly (I:C) in TransIT-X2 Dynamic Delivery System or treated (no transfection) with 1 ng/mL of recombinant human IFN-β for 12 h. WT cells were mock transfected to provide a background control. The cells were washed with phosphate buffered saline (PBS), lyzed in 100 μL of 1× lysis buffer (Promega, Cat# E397A), and lysates were cleared by brief centrifugation. 20 μL of lysates were mixed with 100 μL of luciferase assay reagent (Promega, Cat# E1501) and luminescence was measured by a BioTek Cytation I plate reader. The final result was expressed as a ratio of (luminescence value of knockout minus background) to (luminescence value of WT minus background value) (Supplementary Data 2).

## Type I IFN bioassay

The culture media of mock- or poly (I:C)-treated human cells (appropriately diluted when necessary) were applied to 2fTGH-ISRE-Luc cells at ~70% confluence for 12 h. Six different concentrations of recombinant human IFN-β (25, 50, 100, 200, 400 pg/mL) were included to construct a standard curve. Luminescence was quantified as above. The final result was presented as a raw luminescence value minus mock or converted into pg/mL with the standard curve.

## Dual luciferase assay

HEK293T cells in 24-well plate at ~70% confluence were transfected with 50 ng of Actin promoter-driven renilla luciferase reporter plasmid (internal control), 100 ng of pGL3-ISRE or pGL3-IFNB1 firefly luciferase reporter plasmid, and a gene-of-interest (e.g., UBR5) expression plasmid as appropriate. Twenty-four hours after transfection, luciferase activity was measured using a Dual-Luciferase® Reporter Assay System (Promega, Cat# E1910) according to the manufacturer's instructions. In some cases, cells were transfected with RLR ligands for desired times followed by measurement of luciferase activity. Cells transfected with only two reporter plasmids provided background levels of two different luminescence. The firefly luminescence was normalized with renilla luminescence as a ratio of (firefly luminescence minus background) to (renilla luminescence minus background). The final result was expressed as a ratio of normalized luminescence of a knockout to WT.

## Purification of total cellular RNA, and RT-qPCR

Approximately thirty mg of animal tissues, $25\,\mu L$ of whole blood, and up to $1 \times 10^6$ culture cells were collected in $350\,\mu L$ of lysis buffer (RNApure Tissue & Cell Kit, CoWin Biosciences, Cambridge MA, United States). Soft tissues were homogenized using an electric pellet pestle (Kimble Chase LLC, United States). RNA was extracted according to the product manual. Reverse transcription of RNA into complementary DNA (cDNA) was performed using the PrimeScript™ RT reagent Kit (TaKaRa Bio, Inc, Cat# RR037A). Quantitative PCR (qPCR) was performed with gene-specific primers and iTaq Universal SYBR Green Supermix (BioRad, Cat# 1725124). Results were calculated using the $-\Delta\Delta Ct$ method and a housekeeping gene (ACTB) as an internal control. The qPCR primers and probes for immune genes were reported in our previous studies[16,53].

## Enzyme-Linked ImmunoSorbent Assay (ELISA)

The type I IFN concentrations in cell culture supernatants and murine sera were quantified by a bioluminescent mouse IFN-β ELISA kit 2.0 (InvivoGen, Cat# luex-mifnbv2), a human IFN-β DuoSet ELISA kit (R&D, Cat# DY814-05) or a mouse IFN-α ELISA Kit (Invitrogen, Cat# BMS6027). A LEGENDplex™ bead-based multiplex ELISA kit (Biolegend, San Diego, CA, USA) was employed to quantify multiple serum cytokine concentrations. The procedures were exactly the same as in the product manuals. For the LEGENDplex assay, briefly, ($25\,\mu L$) of samples or standards were mixed with antibody-coated premix microbeads in a filter-bottom microplate and incubated at room temperature for 2 h with vigorous shaking at $500 \times g$. After removal of unbound analytes and two washes, $25\,\mu L$ of detection antibody were added to each well, and the plate was incubated at room temperature for 1 h with vigorous shaking at $500 \times g$. $25\,\mu L$ of SA-PE reagent was then added directly to each well, and the plate was incubated at room temperature for 30 min with vigorous shaking at $500 \times g$. The beads were washed twice with wash buffer, re-suspended in 1× wash buffer, then transferred to a 96-well microplate. The beads were run through a BIORAD ZE5 and the concentration of each analyte was calculated with the standards using the LEGENDPlex data analysis software.

## Immunoblotting

Standard sodium dodecyl sulfate-polyacrylamide gel electrophoresis (SDS-PAGE), Western blotting, and an enhanced chemiluminescent (ECL) substrate (ThermoFisher, Cat# 32106) were applied. In some cases, such as K63-Ub, an ultra-sensitive ECL substrate for low-femtogram-level detection (ThermoFisher, Cat# 34095) was used. An anti-rabbit IgG, HRP-linked secondary antibody (Cell Signaling Technology, Cat#7074) and anti-mouse IgG, HRP-linked secondary antibody (Cat#7076) were used at a dilution of 1:5000 or 1:10000 in 5% milk or bovine serum albumin (BSA). The density of immunoblots was quantified by Image J.

## Immunoprecipitation (IP)

HEK293T cells were transfected with expression plasmids using TransIT-X2 Dynamic Delivery System for 24 h. In some cases, cells were then transfected with a RLR agonist or infected with a virus. Whole-cell extracts were prepared from the transfected/infected cells in a lysis buffer (150 mM NaCl, 50 mM Tris, pH 7.5, 1 mM EDTA, 0.5% NP40, 10% glycerol, protease inhibitors) and were incubated with $20\,\mu L$ of anti-FLAG or Myc magnetic beads (50% v/v) (Cat # A36797, 88842, ThermoFisher). overnight at 4 °C. The beads were washed 5 times lysis buffer and bound proteins were eluted with a 3x FLAG peptide.

For immunoprecipitation of endogenous proteins, cells (a 10-cm culture dish, 100% confluent) were lysed in 1.2 mL of lysis buffer. The lysate was cleared by centrifugation at $12,000 \times g$ for 10 min at 4 °C. $2\,\mu g$ of a mouse anti-TRIM28 antibody (BioLegend, Cat # 619302) or a control mouse IgG were added into $800\,\mu L$ of cell lysate and incubated with gentle agitation at 4 °C overnight. On the second day, $25\,\mu L$ of Pierce Protein A/G magnetic beads (0.25 mg) (Thermo Scientific™ Pierce™, Cat# 88803) were washed and added to the cell lysate. The reaction mix was incubated with gentle agitation at room temperature for 1 h. The beads were collected with a magnetic stand and washed 4 times. Bound proteins were eluted in $80\,\mu L$ of 1× SDS sample buffer at 95 °C for 10 min or 0.1 M pH2.5 glycine.

## Chromatin immunoprecipitation (ChIP) for PCR and ChIP-seq

We used a ChIP-IT® Express Enzymatic Shearing Kit (ActiveMotif, Cat #53035) combining with a ChIP assay kit (Millipore-Sigma, Cat #17-295) and a ChIP DNA Clean & Concentrator kit (ZYMO research, Cat #D5205) to perform the ChIP experiments. Briefly, ~$1 \times 10^7$ cells in a 10-cm cell culture dish were infected with VSV (MOI = 1) for 8 h at 37 °C, 5% $CO_2$. The cells were cross-linked with 1% formaldehyde for 10 min at 37 °C, washed twice with ice cold PBS containing protease inhibitors (Millipore, Cat #539134), then scraped off. The cells were centrifuged for 5 min at $1000 \times g$ at 4 °C and the pellet was resuspended in 1 mL of ice-cold Lysis buffer (ActiveMotif, Cat #53035) supplemented with $5\,\mu L$ Proteinase Inhibitor Cocktail (PIC) and $5\,\mu L$ PMSF (100 mM), vortexed briefly and incubated on ice for 30 min, and then Dounce homogenized with 20 strokes to aid in nuclei release. The nuclei were pelleted for 10 min at $5000 \times g$ at 4 °C, and then resuspended in $350\,\mu L$ Digestion Buffer containing $1.75\,\mu L$ PIC and $1.75\,\mu L$ PMSF (ActiveMotif, Cat #53035), followed by incubating for 5 min at 37 °C. The pre-warmed suspension was mixed with $17\,\mu L$ of Enzymatic Shearing Cocktail (200U/ml), and then incubated at 37 °C for 15 min. $7\,\mu L$ of ice-cold 0.5 M EDTA was then added to stop the reaction and chilled on ice for 10 min. The sheared chromatin in the supernatant was collected by centrifuging for 10 min at $15,000 \times g$ at 4 °C to remove the pellet. $50\,\mu L$ of each sheared chromatin sample was used to assess the efficiency of DNA shearing and concentration after performing reverse-crosslink and DNA extraction. The rest of the sheared chromatin (~$300\,\mu L$) was subjected to immunoprecipitation following the protocol of ChIP assay kit (Millipore-Sigma, Cat #17-295). In brief, the sheared chromatin was diluted to 2 mL in ChIP dilution buffer (Millipore, Cat #20-153) containing protease inhibitors. A portion (~$20\,\mu L$) of the diluted cell lysate was reverse crosslinked with $1\,\mu L$ of 5 M NaCl at 65 °C for 4 h. This was used to quantitate the amount of input DNA for PCR. The remaining crosslinked lysate was precleared with $75\,\mu L$ of Protein A agarose/salmon sperm DNA (Cat # 16-157 C) with agitation for 30 min at 4 °C. After brief centrifugation, the supernatant fraction was subjected to immunoprecipitation with a mouse anti-TRIM28 antibody ($8\,\mu g$/reaction, BioLegend, Cat # 619302) rotating overnight at 4 °C. Protein A agarose/salmon sperm DNA ($60\,\mu L$) was then added and incubated for 1 h at 4 °C. After gentle centrifugation ($1000 \times g$, 4 °C, ~1 min), the agarose beads were washed for 4 min rotating in a series of wash buffer (low salt, high salt, LiCl immune wash buffer and TE buffer). The protein A/antibody/DNA complex was eluted twice using $250\,\mu L$ of fresh elution buffer (1%SDS, 0.1 M NaHCO3) at room

temperature for 15 min, the total eluates (~500 μL) were reversely crosslinked by adding 20 μL of 5 M NaCl and heating at 65 °C for 4 h. The eluates were then incubated with 10 μL of 0.5 M EDTA, 20 μL of 1 M Tris-HCl, pH 6.5 and 1 μL of 20 mg/mL proteinase K for one hour at 45 °C. Ultra-pure DNA was recovered and purified using a ChIP DNA Clean & Concentrator kit (ZYMO research, Cat #D5205), and used for PCR and ChIP-seq. The primers used for ChIP-qPCR were: Forward 5′-AAAAGGTGGCGTCTCCCTGA-3′, Reverse 5′-TCGATTCTCTTCTCA-GGACTTTGT-3′ for *IFIH1*; Forward 5′-CGCTAGTTGCACTTTCGATTT-3′, Reverse 5′-AAAGC-AGGGATTTTCCGAGG-3′ for *DDX58*; Forward 5′-TACTAGCGGTTTTACGGGCG-3′, Reverse 5′-GGCTGCGGGCTCAATT TATAG-3′ for *GAPDH*. The samples were then performed ChIP-seq at Yale Center for Genome Analysis. ChIP-seq reads were aligned to human reference assembly GRCh38 using Bowtie2 v2.3.5.1 aligner[57]. Peaks were identified by comparing the ChIP and input data using MACS2 v2.1.2[58]. The peaks were annotated by their proximity to transcription start site of the corresponding genes, ranging from −3 to +3 kb in the sample[59].

## IP-UPLC-MS/MS analysis

UBR5 or TRIM28 were overexpressed in WT and *UBR5*$^{-/-}$ HEK293T cells by transfection of FLAG-UBR5 (Addgene #37188) or FLAG-TRIM28 (Addgene #124960) plasmids for 24 h and then treated as indicated. The vector transfection in WT was included as negative control. Cells were then washed by cold PBS and lysed using cytoplasmic protein extraction buffer (10 mM HEPES pH8.0, 1.5 mM MgCl2, 10 mM KCl, 0.05% NP40, 0.1 mM EDTA, 1X proteinase inhibitor in di-water) on ice for 30 min. The lysates were centrifuged at 4000 g, 4 °C for 10 min. The supernatants were collected as cytoplasmic extractions and pellets were resuspended in nuclear lysis buffer (10 mM HEPES pH8.0, 1.5 mM MgCl2, 400 mM NaCl, 0.1 mM EDTA, 20% Glycerol, 1X proteinase inhibitor in di-water) and incubated on ice for 30 min, a 30-gauge syringe was used to break up nuclear. Supernatants were collected as nuclear extractions after centrifuge at 13,000 × g, 4 °C for 15 min. Extractions from cytoplasm and nuclear were then mixed thoroughly and subjected to immunoprecipitation (IP) using anti-Flag-M2 magnetic beads. The pulldown quality control was performed by running a Coomassie stained gel. Samples from IP were then subjected to Ultra performance liquid chromatography-tandem mass spectrometer (UPLC-MS/MS) at the UConn Proteomics & Metabolomics Facility.

## Site-directed mutagenesis of TRIM28

The specific ubiquitination sites of TRIM28 were identified by IP-MS (Supplementary Data 6). Site mutations of lysine to arginine of TRIM28 were performed by using a QuikChange II Site-Directed Mutagenesis Kit (Agilent Tech, Cat#200523) and sequenced for verification. Briefly, FLAG-TRIM28 plasmid template was isolated from a dam+ MultiShot™ StripWell Stbl3™ Competent Cell strain. Subsequent PCR amplification with specific mutagenic primers were setup to cycling reaction of 95 °C for 30 s, (95 °C for 30 s, 55 °C for 1 min, 68 °C for 7 min) ×16 cycles, the reaction was then placed on ice for 2 min to cool down to ≤37 °C. The amplification products were then digested by adding 1 μL of the Dpn I restriction enzyme (10 U/μL) and incubating at 37 °C for 1 h. 1 μL of the Dpn I-treated DNA were picked up for transformation of XL1-Blue super-competent cells and mutant plasmids were extracted using a PureLink™ Quick Plasmid Miniprep Kit (Invitrogen, Cat# K210011) followed by sequencing at the GENEWIZ® company. Plasmids containing the desired mutations were then transfected into target cells for expression of site mutant TRIM28 according to designed experiments. Primers for the Site-direct mutagenesis of TRIM28 and verification sequence are in the Supplementary Data 7.

## RNA-sequencing

WT, *UBR5*$^{-/-}$ and *TRIM28*$^{-/-}$ cells were stimulated with poly (I:C) for 12 h or recombinant human IFN-β for 6 h. RNA isolation from cells were performed using an RNA purification kit (Invitrogen, Cat# 12183018A). RNA-seq library preparation and sequencing were performed by GENE-WIZ/Azenta NGS service. Sequence reads were trimmed to remove possible adapter sequences and nucleotides with poor quality using fastp v.0.23.1[60]. UMI-based deduplication was performed using fastp v.0.23.1. Trimmed and deduplicated reads were then mapped to human reference assembly GRCh38 using the STAR aligner v.2.5.2b[61] to generate BAM files. Gene hit counts were calculated by using feature counts from the Subread package v.1.5.2[62]. DESeq2 was used to perform differential expression analysis between the defined groups of samples[63]. The Wald test was used to generate p-values and Log2 fold changes.

## Immunofluorescence microscopy

Cells were cultured on a slide and fixed with 4% paraformaldehyde. The cells were sequentially permeabilized with 0.25% Triton X-100 in PBS, blocked with a blocking buffer (10% goat serum, 0.1% bovine serum albumin, 0.25% Triton X-100 in PBS) at room temperature for 1 h, briefly washed with PBS, incubated with primary antibodies at 4 °C overnight, washed with PBS briefly, and then incubated with Alexa Fluor 488/594-conjugated secondary IgG (1:200, ThermoFisher) for 1 h at room temperature. The nuclei were counter-stained with DAPI (4′,6-diamidino-2-phenylindole). Images were acquired with a Zeiss 880 confocal microscope (objective ×40, ×63 oil).

## Graphing and statistical analyses

The sample sizes for the animal experiments were estimated according to our prior experience in similar experiments and power analysis calculations (http://isogenic.info/html/power_analysis.html). All the animals were included for analysis, and no method of randomization was applied. A GraphPad Prism software was used for graphing and statistical analysis. Survival curves were analyzed using a log-rank (Mantel−Cox) test. A standard two-tailed, unpaired Student's *t* test or non−parametric/parametric Mann−Whitney U test was applied to a data set depending on its data distribution. The One-Way ANOVA test was applied to simultaneously comparing multiple groups. *p* values ≤ 0.05 were considered significant. The sample sizes (biological replicates), statistical tests, and *p* values were specified in the figure legends.

## Reporting summary

Further information on research design is available in the Nature Portfolio Reporting Summary linked to this article.

# Data availability

The datasets generated during and/or analyzed during the current study are in the supplemental materials. Source data are provided with this paper. The raw data and processed data for RNA-seq and ChIP-seq are available at National Center for Biotechnology Information GEO (GSE245604). Source data are provided with this paper.

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

## Acknowledgements

This work was supported by National Institutes of Health grants R01AI132526 and R21AI177623 to P.W. and a UConn Health startup fund to P.W.

## Author contributions

D.Y. performed most experimental procedures, acquired, and analyzed the data. T.G., A.G.H., J.G.C., B.J., and M.W. assisted D.Y. with some experimental procedures and/or provided technical support. J.X. performed bioinformatic analysis. R.E.H. provided the UBR5-loxP mouse model. C.C., H.W., A.T.V., G.C. and Y.W. helped with result discussion, provided key reagents or instruments. D.Y. and P.W. conceived, designed the study, and wrote the manuscript. All the authors reviewed and/or modified the manuscript.

## Competing interests

The authors declare no competing interests.
