## [Peer Review File · Nature Communications]

UBR5 promotes antiviral immunity by disengaging the transcriptional brake on RIG-I like receptorsREVIEWER COMMENTS

Reviewer #1 (Remarks to the Author):

In their study, Yang et. al. aimed to identify new ubiquitin E3 ligases that could regulate innate antiviral immune signaling pathways. They used a combination of luciferase reporter models and CRISPR methods to screen for potential candidates and discovered UBR5, a ubiquitin protein ligase E3 component n-recognin 5, which has a significant impact on the responses against RLR ligands and RNA viruses, and to a lesser extent, on cGAS ligands. Through in vitro and in vivo experiments, the authors showed that UBR5 affects host defense against both RNA and DNA viral infections, independently of Interferon signaling, using the EMCV and HSV-1 infection models. Mechanistically, the authors found that UBR5 interacts with TRIM28 and promotes RLR transcription by affecting TRIM28 protein modification through the regulation of ubiquitination and SUMOylation.

Overall, although the study presents potentially intriguing findings on UBR5 and its role in regulating the RLR expression, the authors fail to show a convincing explanation of the mechanism of UBR5 in relation to TRIM28 functions. Without transcriptome-side analysis, it is not possible for the authors to claim that UBR5 specifically target RIG-I and MDA5 for their transcription. If this is the case, the authors should analyze the reason why UBR5 is so specific, given that TRIM28 is a chromatin regulator that can control gene expression more broadly. Indeed, a previous report showed that TRIM28/KAP1 interacts with IRF5 to control inflammatory gene expression. The other issue is that it is possible that UBR5 affects TRIM28-mediated epigenetic changes instead of direct transcriptional activity. The authors need to perform RNA sequencing analysis to decipher how difference the impact of UBR5 and TRIM28 alone on impacting RLR expression.

Specific comments:

1. The authors should analyze the transcriptome profile of UBR5 deficient cells with/without RLR ligands to examine whether UBR5 specifically target RIG-I and MDA5. Given these RLRs are IFN-inducible, the authors can also consider stimulating cells type I IFNs to check transcriptome.
2. Although the authors identified TRIM28 as the UBR5 binding partner and the target for K63-linked ubiquitination, it is expected that there should be many more UBR5 targets. Thus the authors can perform experiments to isolate UBR5-associating proteins in a more unbiased way, such as IP-MS analysis.
3. It is also important to examine the overlap of differentially regulated genes in UBR5 and TRIM28 deficient cells, if they are controlling the expression of RLRs specifically.
4. In Figures 5e and 5f, a better resolution and quantification of MDA5 and RIG-I protein expression could strengthen the evidence of UBR5's effect on their expression levels. The current result does not suggest that overexpression of UBR5 increase RIG-I and MDA-5 protein levels significantly to the same extent as UBR deficient condition. Also, the immunoblots in Figure 7, especially Figure 7d, need quantification.
5. In Figure 7b, the data showing the decreased levels of TRIM28 ubiquitination in UBR5 deficient cells are not convincing. Even Flag-TRIM28 expression seems to be decreased in UBR5 deficient cells expressing HA-K63Ub.
6. The relationship between the decreased ubiquitination and increased SUMOylation is not convincingly shown. The authors need to identify the sites of TRIM28 for ubiquitination and SUMOylation.
7. In Figure 7e, the authors show that the recruitment of TRIM28 to MDA5 and RIG-I gene loci was increased in UBR5 deficient cells. Again, the authors can perform ChIP-sequence analysis to investigate the specificity for the control of TRIM28 on RLR alleles.
8. In this study, the authors used the terms "UBR5+/+" and "UBR5-/-" to distinguish between

different cell types. However, some readers might find this terminology confusing as they might mistakenly assume that "UBR5^{-/-}" primary cells are from conventional UBR5 knockout mice such as macrophages or MEFs, even though the authors explained in the text that these mice were derived from inducible KO models.

Reviewer #2 (Remarks to the Author):

In this manuscript, Yang et al reported the identification of an E3 ligase UBR5, which might be involved in RLR signaling. UBR5 regulates RLR transcription through inhibition on TRIM28. UBR5 plays a role in both the basal expression and induction of RLR upon virus infection, and is required for antiviral response. Though the data is largely reasonable, there are a few concerns to be addressed.

1. Downregulation of RLR by the absence of UBR5 was shown. However, the activation or signaling status of major players in RLR pathway need to be investigated with or without stimulation.
2. To rule out the possibility that UBR5 might be required for the antiviral function of other players in RLR pathway, overexpression of RIG-I(N), MAVS and TBK1 etc., should be performed and investigated in UBR5 KO cells.
3. mRNA level of IFN β is not correlated with its protein level (Fig 2m and n, Fig 2p and q). An explanation is needed for this discrepancy.
4. In Fig 3h, HSV-1 load is much higher in UBR5^{-/-} cells, which should not have been described as modest. In Fig 3i, IFN induction is decreased in UBR5^{-/-} cells, consistent with HSV-1 proliferation. These data suggest that UBR5 might be involved in cGAS pathway.
5. In Fig 5c elevation of basal level of RLR in UBR5^{-/-} cells is very trivial and neglectable, undermining the major conclusion of manuscript.
6. In Fig 7c, WB with anti-Ub and sumo antibodies reveal singular and sharp bands, while in Fig 7d, similar WB reveal smear bands. How can these be?
7. How big is UBR5? Its size shown in Fig 6f seems to be different from other WBs.

Reviewer #3 (Remarks to the Author):

RLR is an RNA sensor essential for the initiation of antiviral immune responses against RNA viruses, and ubiquitination is known to be important in its transcription. In this study, the authors found that UBR5 is a positive regulator of RLR transcription; UBR5 deficiency reduces the antiviral immune response to RNA viruses and increases viral replication UBR5 binds to and ubiquitinates the lysine 63 of TRIM28 and inhibits TRIM28 SUMOylation, thereby upregulating RLR expression and promoting antiviral immune responses. The function of UBR5 as a positive regulator and TRIM28 as a negative regulator in RLR expression is very interesting. The proposal that the UBR5-TRIM28 axis functions as a rheostat is also acceptable.

Some comments to improve the manuscript.

Is the citation in line 59 appropriate? This is a statement that focuses on MDA5.

The reviewer could not immediately find from the two cited references what the source of the reference to the A946T variant of MDA5 in line 68 is.

Please unify whether the style of references is by number or name.

The text on lines 163-172 does not match the METHOD. Did the female parent give tamoxifen to make the fetus Ubr5-KO?

The citation on line 166 is a duplicate.

Hyphenate Calu3 in Figure 5a.

Ext Fig. 6b with "kDa".

The abbreviation for TRIM28 in line 260 has already appeared in line 90.

In lines 270-182, the authors analyze the binding between UBR5 and TRIM28. In light of the data in Fig. 7d, am I correct in understanding that the binding of both endogenous proteins in the unstimulated state is weak?

The "promoter" should be appended to IFIH1 and DDX5 in Figure 7e.

Lines 359-362 duplicate what is stated from line 332.

Flag in Ext Fig. 9b should be capitalized.

Point-by-point response to the reviewers' comments

Reviewer #1 (Remarks to the Author):

In their study, Yang et. al. aimed to identify new ubiquitin E3 ligases that could regulate innate antiviral immune signaling pathways. They used a combination of luciferase reporter models and CRISPR methods to screen for potential candidates and discovered UBR5, a ubiquitin protein ligase E3 component n-recognin 5, which has a significant impact on the responses against RLR ligands and RNA viruses, and to a lesser extent, on cGAS ligands. Through in vitro and in vivo experiments, the authors showed that UBR5 affects host defense against both RNA and DNA viral infections, independently of Interferon signaling, using the EMCV and HSV-1 infection models. Mechanistically, the authors found that UBR5 interacts with TRIM28 and promotes RLR transcription by affecting TRIM28 protein modification through the regulation of ubiquitination and SUMOylation.

Overall, although the study presents potentially intriguing findings on UBR5 and its role in regulating the RLR expression, the authors fail to show a convincing explanation of the mechanism of UBR5 in relation to TRIM28 functions. Without transcriptome-side analysis, it is not possible for the authors to claim that UBR5 specifically target RIG-I and MDA5 for their transcription. If this is the case, the authors should analyze the reason why UBR5 is so specific, given that TRIM28 is a chromatin regulator that can control gene expression more broadly. Indeed, a previous report showed that TRIM28/KAP1 interacts with IRF5 to control inflammatory gene expression. The other issue is that it is possible that UBR5 affects TRIM28-mediated epigenetic changes instead of direct transcriptional activity. The authors need to perform RNA sequencing analysis to decipher how difference the impact of UBR5 and TRIM28 alone on impacting RLR expression.

Response: We appreciate this reviewer's insightful critiques and suggestions. We totally agree that UBR5-TRIM28 regulates gene expression epigenetically, thus have modified relevant terms in the text. Per the reviewer's recommendation, we performed RNA sequencing analysis, ChIP-seq analysis and IP-UPLC-MS/MS analysis to improve mechanistic insights. We also identified the specific lysine residue within TRIM28 modified by UBR5.

TRIM28 was shown to interact with IRF5 to control inflammatory gene expression in macrophages. However, IRF5 is less relevant to RLR signaling and type I IFNs. Nonetheless, we discussed this and included relevant citations (Page 17-18, Line 433-438).

Specific comments:

1.1. The authors should analyze the transcriptome profile of UBR5 deficient cells with/without RLR ligands to examine whether UBR5 specifically target RIG-I and MDA5. Given these RLRs are IFN-inducible, the authors can also consider stimulating cells type I IFNs to check transcriptome.

Response 1.1: We performed RNA-seq analyses of both WT and *UBR5*^{-/-} HEK293 cells treated with/without poly (I:C). In untreated cells, 553 genes were downregulated (Log₂FC<-1), including *IFIH1* (Log₂FC: -1.74, *p*<0.05); *DDX58* was modestly downregulated (Log₂FC: -0.62, *p*<0.05) (**Fig.8a**). These results were largely consistent with the PCR data (**Fig.5g**). DEGs were enriched in RNA Polymerase III Chain Elongation (**Suppl Table 9**). Intriguingly, after poly (I:C) stimulation, downregulated genes (*UBR5*^{-/-} vs WT) (Log₂FC<-1, *p*<0.05) were enriched primarily in IFN-I signaling and TRAF3-dependent IRF activation pathways including *IFNB1*,

DDX58 and *IFIH1* etc. (**Fig.8b-d**). We described (Page 15, Line 360-369) and discussed (Page 17, Line 419-424) the new data.

There was no difference in RLR and other conventional ISG expression between WT and *UBR5*^{-/-} cells treated with recombinant human IFN- β (**Extended Data Fig. 11b, Suppl. Table 10**), consistent with the primary CRISPR screening results (**Fig.1d**) showing that IFN- β -induced ISRE-Luc was no different between WT and *UBR5*^{-/-} cells. These results suggest that UBR5 is dispensable for the IFNAR1-JAK-STAT1/2 pathway. However, if UBR5 promotes RLR chromatin accessibility by disengaging TRIM28 from the chromatin, one would anticipate reduced RLR expression in *UBR5*^{-/-} cells during IFN- β treatment. One possibility is that other TRIM28-regulating mechanisms may be predominant over UBR5 during IFN- β treatment. In addition to ubiquitination, TRIM28 is regulated by phosphorylation. Ataxia-Telangiectasia Mutated (ATM) kinase, a member of nuclear phosphatidylinositol-3 kinase-like (PIKK) family, mediates phosphorylation of TRIM28, resulting in chromatin relaxation (PMID: 28851455). Notably, type I IFNs drive ATM-dependent TRIM28 phosphorylation (PMID: 28362262). Thus, phosphorylation may be the dominant regulation of TRIM28 downstream of IFNAR-JAK-STAT1/2. To see if TRIM28 targets individual ISGs or the whole IFNAR-JAK-STAT1/2 pathway, we looked into the upregulated DEGs in *TRIM28*^{-/-} vs WT after IFN- β stimulation. In WT HEK293 cells, treatment with recombinant IFN- β for 6 hr upregulated 331 genes, which are collectively termed ISGs, including well-characterized conventional ISGs, *SOCS1*, *MX1/2*, *RSAD2*, *OAS1/2/3*, *IFIT1/2/3*, *IFIH1*, *DDX58*, *ISG15/20*, *IFITM1/2* etc. (**Extended Data Fig. 11a, Suppl. Table 11**). Following IFN- β stimulation, 292 genes (Log₂FC>1, *p*<0.05) including *DDX58* were upregulated in *TRIM28*^{-/-} compared to WT cells. The Log₂FC in *IFIH1* was 0.8, *p*<0.05. Notably, in *TRIM28*^{-/-} cells, a small number (22) of conventional ISGs including *DDX58*, *MX1/2*, *OAS1/2/3*, and *IFIT2/3* were upregulated, suggesting that TRIM28 selectively targets individual ISGs rather than the whole IFNAR-JAK-STAT1/2 pathway (**Suppl. Table 11**). Intriguingly, *Ddx58*, *Ifih1*, *Mx1*, *Ifit1* were also among the 6 genes directly targeted by TRIM28 in murine peritoneal macrophages (PMID: 34497149). We described (Page 15, Line 378-390) and discussed (Page 17, Line 422-424, Line 427-433) these new results.

Of note, although the IFNAR-JAK-STAT1/2 pathway induces significant RLR expression, initial RLR levels determine the magnitude of IFN-I and subsequent ISG expression. Moreover, RLR expression can be induced by IRF3 in a cytokine-independent manner (PMID: 17475649). Therefore, UBR5 regulates early RLR expression and signaling that is critical for the control of viral replication.

1.2. *Although the authors identified TRIM28 as the UBR5 binding partner and the target for K63-linked ubiquitination, it is expected that there should be many more UBR5 targets. Thus the authors can perform experiments to isolate UBR5-associating proteins in a more unbiased way, such as IP-MS analysis.*

Response 1.2: Per the reviewer's recommendation, we transiently expressed UBR5-FLAG in HEK293T cells, and then performed immunoprecipitation with an anti-FLAG IgG, identified UBR5-bound proteins by UPLC-MS/MS. Fifty-two proteins were significantly enriched by FLAG-UBR5 compared to the vector alone. Seventeen proteins including TRIM28 were previously reported (**Fig. 6e**) (Page 11, Line 271-276).

1.3 *It is also important to examine the overlap of differentially regulated genes in UBR5 and TRIM28 deficient cells, if they are controlling the expression of RLRs specifically.*

Response 1.3: We did RNA-seq to analyze the overlapping differentially expressed genes (DEGs) in UBR5 and TRIM28 deficient cells stimulated with poly (I:C). Considering the opposite role of UBR5 and TRIM28 in RLR signaling, we compared the downregulated genes from *UBR5*^{-/-} with the upregulated genes from *TRIM28*^{-/-} cells. There were 42 overlapping DEGs, among which 9 genes were related to antiviral immunity including *DDX58*, 23 genes were primarily involved in RNA Polymerase II transcription including 22 Zinc Finger (ZNF) proteins and

BATF2 (**Fig. 8e**). Surprisingly, *IFIH1* was not significantly upregulated in *TRIM28*^{-/-} compared to WT cells treated with poly (I:C) for 12 hr, although its encoded protein MDA5 was obviously upregulated in *TRIM28*^{-/-} cells treated with either poly(I:C) or VSV for 12 hr (**Fig.6a**). This discrepancy is probably due to the induction of RLR mRNA by poly (I:C) for 12 hr in WT and *TRIM28*^{-/-} was close to their plateau. Indeed, following poly (I:C) stimulation for 12 hr, the Log2 fold changes of RLRs in *UBR5*^{-/-} cells (-1.96 and -2.1) were more significant than those in *TRIM28*^{-/-} (1 and 0). Therefore, 12 hr may be ideal for identification of downregulated DEGs, but an earlier time point may be better for upregulated DEGs by RNA-seq. However, we believe it is more appropriate to identify overlapping DEGs between two knockouts stimulated with the same condition (cell type, concentration, time). Notably, both *DDX58* and *IFIH1* were upregulated in *TRIM28*^{-/-} vs WT (Log2FC: 1.02 and 0.81) following IFN-β stimulation for 6 hr, suggesting that they are TRIM28 targets (Response 1.1, PMID: 34497149). Moreover, ChIP-seq shows that TRIM28 binding to both *DDX58* and *IFIH1* DNA increased in *UBR5*^{-/-} cells (Response 1.7, **Fig. 7k**). All these results suggest that UBR5-TRIM28 regulates *DDX58* and *IFIH1* expression epigenetically. We described these new results (Page 15, Line 369-377).

1.4. *In Figures 5e and 5f, a better resolution and quantification of MDA5 and RIG-I protein expression could strengthen the evidence of UBR5's effect on their expression levels. The current result does not suggest that overexpression of UBR5 increase RIG-I and MDA-5 protein levels significantly to the same extent as UBR deficient condition. Also, the immunoblots in Figure 7, especially Figure 7d, need quantification.*

Response 1.4: We improved the quality of all these immunoblots and quantified their band density (**Fig. 5e, f, and Fig. 7**)

1.5. *In Figure 7b, the data showing the decreased levels of TRIM28 ubiquitination in UBR5 deficient cells are not convincing. Even Flag-TRIM28 expression seems to be decreased in UBR5 deficient cells expressing HA-K63Ub.*

Response 1.5: The difference in ubiquitinated TRIM28 is greater than that in total FLAG-TRIM28 input. But indeed, there is a moderate decrease in TRIM28 input in UBR5-deficient cells. We therefore updated the immunoblots (**Fig.7b**).

1.6. *The relationship between the decreased ubiquitination and increased SUMOylation is not convincingly shown. The authors need to identify the sites of TRIM28 for ubiquitination and SUMOylation.*

Response 1.6: To identify the specific ubiquitin acceptor lysine residue of TRIM28, we expressed FLAG-tagged TRIM28 in WT and *UBR5*^{-/-} cells with or without Poly (I:C) treatment, performed immunoprecipitation, identify differentially ubiquitinated lysine residues by UPLC-MS/MS. We found that K507 ubiquitination increased after poly (I:C) in WT cells, however, was unchanged in *UBR5*^{-/-} cells, indicating that K507 is the UBR5 target (**Suppl. Table 6, Fig. 7f**). Next, we constructed a K507R mutant TRIM28 plasmid and confirmed that UBR5-mediated ubiquitination of TRIM28 was lost, and in contrary, K507R TRIM28 SUMOylation increased. We also included K779R mutant TRIM28 as a SUMOylation-deficient control (PMID: 18082607) (**Fig. 7g, h**).

Though not the key SUMOylation site (PMID: 18082607), K507 lies in the heterochromatin protein 1 (HP1)-binding domain (HP1BD) (aa 483-510) (PMID: 10330177) and is conserved in humans and rodents. Of note, HP1BD is also critical for the nuclear targeting, repression function and SUMOylation of TRIM28 (PMID: 18082607). Therefore, UBR5-mediated ubiquitination of K507 may likely affect TRIM28 functionality. Indeed, UBR5 does inhibit not only TRIM28 SUMOylation but also binding to the RLR promoters (**Fig.7**). We described (Page 13-14, Line 324-338) and discussed these new results (Page 18, Line 455-460).

Penghua Wang, Ph.D., Assoc. Prof.
Department of Immunology, School of Medicine
The University of Connecticut Health Center
263 Farmington Ave, Farmington, CT 06030
Email: pewang@uchc.edu
Tel: 860-679-6393

1.7. In Figure 7e, the authors show that the recruitment of TRIM28 to MDA5 and RIG-I gene loci was increased in UBR5 deficient cells. Again, the authors can perform ChIP-sequence analysis to investigate the specificity for the control of TRIM28 on RLR alleles.

Response 1.7: We performed the ChIP-seq analysis in WT and UBR5^{-/-} cells treated with/without VSV for 8 hr. Approximately 1933 gene promoters were enriched by over 10-fold enrichment (relative to input whole genome) in UBR5^{-/-} cells, and over 2-fold relative to WT cells. Of note, TRIM28-bound IFIH1/DDX58 promoter DNA was much more in UBR5^{-/-} than that in WT cells. Intriguingly, we noted a similar trend for several IFN-I signaling-related genes including ISG20, CXCL10, MX1 and IFIT1 (**Fig. 7k**), consistent with a recent publication (PMID: 3449714925). However, we did not observe other known viral PRRs (MB21D1, TLR3/7/9, NLRP) or major components of the RLR pathways (MAVS, TBK1, IRF, IFN etc.) (**Suppl. Table 8**). These results suggested that although UBR5 could potentially regulate many genes via TRIM28 during RNA virus infection, it preferably targets RLRs and several ISGs in the context of PRR-IFN signaling. We added these results to the manuscript (Page 14, Line 345-356).

1.8. In this study, the authors used the terms "UBR5^{+/+}" and "UBR5^{-/-}" to distinguish between different cell types. However, some readers might find this terminology confusing as they might mistakenly assume that "UBR5^{-/-}" primary cells are from conventional UBR5 knockout mice such as macrophages or MEFs, even though the authors explained in the text that these mice were derived from inducible KO models.

Response 1.8: Per the reviewer's comments, we changed the terms of "Ubr5^{+/+}" and "Ubr5^{-/-}" to "Ubr5^{WT}" and "Ubr5^{iKO}" to distinguish the inducible knockout (iKO) model.

Reviewer #2 (Remarks to the Author):

In this manuscript, Yang et al reported the identification of an E3 ligase UBR5, which might be involved in RLR signaling. UBR5 regulates RLR transcription through inhibition on TRIM28. UBR5 plays a role in both the basal expression and induction of RLR upon virus infection and is required for antiviral response. Though the data is largely reasonable, there are a few concerns to be addressed.

Response: We thank Reviewer 2 for the constructive critiques.

2.1. Downregulation of RLR by the absence of UBR5 was shown. However, the activation or signaling status of major players in RLR pathway need to be investigated with or without stimulation.

Response 2.1: TBK1 and IRF3 are major components of the RLR pathways and their phosphorylation, i.e., activation, is most commonly examined in the studies of viral PRR signaling. Phosphorylation of TBK1 and IRF3 was reduced in UBR5 deficient cells upon poly (I:C) stimulation (**Fig. 5c**). We added these results to the manuscript (Page 10, Line 240-242).

2.2. To rule out the possibility that UBR5 might be required for the antiviral function of other players in RLR pathway, overexpression of RIG-I(N), MAVS and TBK1 etc., should be performed and investigated in UBR5 KO cells.

Response 2.2: Overexpression of RIG-I, MAVS, TBK1 or IRF3-5D (constitutively active mutant) induced-IFNB1 expression was comparable between WT and UBR5 knockout cells (**Extended Data Fig. 6a**), implying that UBR5 does not directly target them.

Penghua Wang, Ph.D., Assoc. Prof.
Department of Immunology, School of Medicine
The University of Connecticut Health Center
263 Farmington Ave, Farmington, CT 06030
Email: pewang@uchc.edu
Tel: 860-679-6393

2.3. mRNA level of IFN β is not correlated with its protein level (Fig 2m and n, Fig 2p and q). An explanation is needed for this discrepancy.

Response 2.3: mRNA expression proceeds protein expression, and their temporal kinetics are different. We measured *Ifnb1* mRNA expression at early time points, *i.e.*, 6 and 12 hr. For protein quantification, we extended to 24 hr. At 6 hr, even though the induction of the *Ifnb1* mRNA was significant, the IFN- β protein expression was still at its early phase, thus hardly detectable by ELISA. At 12 hr, the differences in both mRNA and protein were significant.

At 24 hr, the difference in IFN- β protein levels between WT and *Ubr5* KO narrowed down. This is likely because that UBR5 regulates the IFN-I response at the early stage of stimulation and/or IFN-I protein expression reaches a plateau at 24 hr when negative regulators kick in (SOCS, A20 etc.). Type I IFNs control viral replication most potently at the early stage.

2.4. In Fig 3h, HSV-1 load is much higher in *UBR5*^{-/-} cells, which should not have been described as modest. In Fig 3i, IFN induction is decreased in *UBR5*^{-/-} cells, consistent with HSV-1 proliferation. These data suggest that *UBR5* might be involved in cGAS pathway.

Response 2.4: We agree. In the beginning of this study, we found that ISD-induced ISRE-Luc reduced by ~50% in *UBR5*^{-/-} cells (**Fig. 1f**), indicating a moderate role of UBR5 in cGAS-STING signaling. Consistently, *Ifnb1* mRNA expression was slightly reduced in *Ubr5* KOMEFs at 2 hr (peak expression), and the serum IFN- β level trended lower in *Ubr5* KO mice at 8 hr (peak level) during HSV-1 infection. However, the HSV-1 load increased by 100% in *Ubr5* KO cells, suggesting that UBR5 could control HSV-1 replication in both IFN-independent and -dependent manners. Indeed, UBR5-TRIM28 could also directly regulate the transcription of a few ISGs, of which, ISG20, MX1, and IFIT1 can directly interfere with the replication of many viruses, including HSV-1 (See discussion on Page 18, Line 439-447).

2.5. In Fig 5c elevation of basal level of RLR in *UBR5*^{-/-} cells is very trivial and neglectable, undermining the major conclusion of manuscript.

Response 2.5: The negligible induction of RLRs in *UBR5*^{-/-} is likely due to weak poly(I:C) stimulation as the induction in WT cells was moderate (2-3-fold) too. We repeated the experiment and updated the immunoblots (**Fig. 5c**).

2.6. In Fig 7c, WB with anti-Ub and sumo antibodies reveal singular and sharp bands, while in Fig 7d, similar WB reveal smear bands. How can these be?

Response 2.6: We think the difference is because of overexpression of FLAG-TRIM28 (**Fig.7c**) and endogenous TRIM28 (**Fig.7d**). Similar patterns of SUMOylation of endogenous TRIM28 and overexpressed FLAG-TRIM28 have been reported earlier (PMID: 18082607, 31391303). SUMOylated endogenous TRIM28 showed multiple smear bands (Fig.i A), while there was one dominant SUMOylated FLAG-TRIM28 band w/wo co-overexpression of His-SUMO2 (Fig.i B,C). This dominant band is likely the monosumoylated form of TRIM28. Similarly, overexpression of FLAG-TRIM28 results in a dominant ubiquitinated band (Fig.ii, PMID: 35406379). We speculate that overexpression of TRIM28 per se may ensue preferably short chain ubiquitination and SUMOylation by an unknown mechanism.

2.7. How big is UBR5? Its size shown in Fig 6f seems to be different from other WBs.

Response 2.7: The molecular weight of UBR5 is 309 kDa, the band is observed at 300 kDa, right above the largest marker band of 250kDa. We made a mistake labelling Fig. 6f and corrected it.

Reviewer #3 (Remarks to the Author):

RLR is an RNA sensor essential for the initiation of antiviral immune responses against RNA viruses, and ubiquitination is known to be important in its transcription. In this study, the authors found that UBR5 is a positive regulator of RLR transcription; UBR5 deficiency reduces the antiviral immune response to RNA viruses and increases viral replication UBR5 binds to and ubiquitinates the lysine 63 of TRIM28 and inhibits TRIM28 SUMOylation, thereby upregulating RLR expression and promoting antiviral immune responses. The function of UBR5 as a positive regulator and TRIM28 as a negative regulator in RLR expression is very interesting. The proposal that the UBR5-TRIM28 axis functions as a rheostat is also acceptable.

Some comments to improve the manuscript.

Response: We appreciate this reviewer for his/her valuable suggestions.

3.1. Is the citation in line 59 appropriate? This is a statement that focuses on MDA5.

Response: We made a mistake of mixing up this citation with the next one and corrected it (Page 3, Line 63, and Line 68).

3.2. The reviewer could not immediately find from the two cited references what the source of the reference to the A946T variant of MDA5 in line 68 is.

Response: Both references (Gateva et al., 2009 and Funabiki et al., 2014) report the association of A946T with SLE. Gateva et al., 2009 reported a highly significant association between SLE and rs1990760 missense allele of *IFIH1*. The single nucleotide polymorphism, rs1990760, results in the amino-acid change (p.A946T) in MDA5.

Fig.i. (A) Immunoprecipitation of SUMOylated proteins from A549 cells and immunoblotted for TRIM28. (B-C) Immunoblots for FLAG-TRIM28 in HEK293T cells overexpressing (B) FLAG-TRIM28 only (C) FLAG-TRIM28 and His-SUMO2. Adapted from Fig.2F and 3F of PMID: 31391303.

Fig.ii. Immunoprecipitation of FLAG-TRIM28 from HEK293T cells overexpressing FLAG-TRIM28 and HA-Ub. Ubiquitination was detected with anti-HA antibody. Adapted from Fig.2A of PMID: 35406379.

Funabiki et al., 2014 further demonstrated that A946T mutation constitutively activated IFN-I signaling and thus involved in SLE. In one word, rs1990760 (SNP) is equivalent to A946T (protein).

3.3. *Please unify whether the style of references is by number or name.*

Response: We formatted all the references to Nat Comm style.

3.4. *The text on lines 163-172 does not match the METHOD. Did the female parent give tamoxifen to make the fetus Ubr5-KO?*

Response 3.4: No. UBR5 is essential for embryonic development. We treated adult female and male ERT2-Cre^{+/-} Ubr5^{flox/flox} mice with tamoxifen or vehicle control, resulting in Ubr5^{KO} or Ubr5^{WT} respectively (per Rev 1 suggestion, we use iKO to differentiate inducible knockout from constitutive knockout). We clarified this in the result section (Page 7, Line 169-170).

3.5. *The citation on line 166 is a duplicate.*

Response 3.5: We're sorry for the mistake and corrected it (Page 7, Line 166).

3.6. *Hyphenate Calu3 in Figure 5a.*

Response 3.6: We're sorry for the mistake and corrected it.

3.7. *Ext Fig. 6b with "kDa".*

Response 3.7: We added "kDa" to the figure.

3.8. *The abbreviation for TRIM28 in line 260 has already appeared in line 90.*

Response 3.8: We removed the unnecessary abbreviation for TRIM28 in this sentence (Page 11, Line 262).

3.9. *In lines 270-182, the authors analyze the binding between UBR5 and TRIM28. In light of the data in Fig. 7d, am I correct in understanding that the binding of both endogenous proteins in the unstimulated state is weak?*

Response 3.9: Yes. Endogenous TRIM28-UBR5 interaction is relatively weak at steady state and was obviously enhanced following VSV infection. This action allows for *DDX58/IFIH1* chromatin relaxation and expression by enhancing TRIM28 ubiquitination and decreasing its SUMOylation.

3.10. *The "promoter" should be appended to IFIH1 and DDX5 in Figure 7e.*

Response 3.10: We appended "promoter" to *IFIH1* and *DDX58* in the figure (**Fig. 7i**).

3.11. *Lines 359-362 duplicate what is stated from line 332.*

Response 3.11: We changed the description of this sentence. (Page 18, Line 449).

3.12. *Flag in Ext Fig. 9b should be capitalized.*

Penghua Wang, Ph.D., Assoc. Prof.
Department of Immunology, School of Medicine
The University of Connecticut Health Center
263 Farmington Ave, Farmington, CT 06030
Email: pewang@uchc.edu
Tel: 860-679-6393

Response 3.12: Corrected.

REVIEWER COMMENTS

Reviewer #1 (Remarks to the Author):

The revised manuscript is substantially improved. Thus, this manuscript is mostly ready for publication.

A minor comment is:

Extended Data Fig. 8c lacks scale bars, although the authors mention the presence of them in the Figure legends.

Reviewer #2 (Remarks to the Author):

2.2. To rule out the possibility that UBR5 might be required for the antiviral function of other players in RLR pathway, overexpression of RIG-I(N), MAVS and TBK1 etc., should be performed and investigated in UBR5 KO cells.

Response 2.2: Overexpression of RIG-I, MAVS, TBK1 or IRF3-5D (constitutively active mutant) induced-IFNB1 expression was comparable between WT and UBR5 knockout cells (Extended Data Fig. 6a), implying that UBR5 does not directly target them.

This reviewer is not satisfied with this response. If overexpression of RIG-I, MAVS, TBK1 or IRF3-5D (constitutively active mutant) induced-IFNB1 expression was comparable between WT and UBR5 knockout cells, How could UBR5 deficiency (UBR5^{-/-}) reduce antiviral immune responses to RNA viruses?

Based on these data, it can be concluded that UBR5 is not involved in RIG-I pathway.

Reviewer #3 (Remarks to the Author):

I am satisfied with the authors comments and changes of the manuscript on addressing my concerns.

Penghua Wang, Ph.D., Assoc. Prof.
Department of Immunology, School of Medicine
The University of Connecticut Health Center
263 Farmington Ave, Farmington, CT 06030
Email: pewang@uchc.edu
Tel: 860-679-6393

Point-by-point response to the reviewers' comments

Reviewer #1 (Remarks to the Author):

The revised manuscript is substantially improved. Thus, this manuscript is mostly ready for publication.

A minor comment is: Extended Data Fig. 8c lacks scale bars, although the authors mention the presence of them in the Figure legends.

Response: We thank this reviewer for the positive feedback to our revisions. For the remaining question, we added the scale bars.

Reviewer #2 (Remarks to the Author):

This reviewer is not satisfied with this response. If overexpression of RIG-I, MAVS, TBK1 or IRF3-5D (constitutively active mutant) induced-IFNB1 expression was comparable between WT and UBR5 knockout cells, How could UBR5 deficiency (UBR5^{-/-}) reduce antiviral immune responses to RNA viruses?

Response: We apologize for not explaining the results clearly in our original response.

Our data and conclusion are that the UBR5-TRIM28 axis epigenetically regulates endogenous RLR transcription. Thus, UBR5 deficiency leads to reduced endogenous RLR expression at steady state and during viral infection, and consequently reduced antiviral immune responses. Mechanistically, TRIM28 is an epigenetic repressor of RLRs transcription by imposing chromatin compaction at the RLR promoter region. UBR5 relaxes TRIM28-imposed chromatin compaction and inhibits TRIM28 binding to RLR promoter DNA by ubiquitinating TRIM28. However, the effect of overexpression of RIG-I or other major components driven by the cytomegalovirus (CMV) promoter from a plasmid is independent of UBR5-TRIM28 regulation (which targets endogenous RLR promoter, but not CMV promoter or endogenous IFNB1 promoter). Overexpressed proteins are much more than their endogenous forms and dictate IFNB1 transcription. Therefore, IFNB1 expression is not impacted by UBR5 deficiency during overexpression of RIG-I-2CARDs, MAVS or TBK1 (**Extended Data Fig.6a**). We apologize that our response "implying that UBR5 does not directly target them" is misleading. Actually, our results demonstrate that UBR5 regulates endogenous RLR transcription but not RLR protein function or downstream component function per se.

To further strengthen the above conclusion, we investigated endogenous RLR expression induced by overexpression of RIG-I 2 CARDs, MAVS or TBK1 in UBR5^{-/-} cells. We transfected individual expression plasmids into WT and UBR5^{-/-} cells for 0, 12 and 24 hrs, and assessed endogenous RIG-I expression by immunoblotting. Recombinant FLAG-RIG-I 2CARDs, Myc-MAVS and GFP-TBK1 protein expression levels were the same between WT and UBR5^{-/-} cells. Although induction of the endogenous RIG-I expression was significant only at 24 hrs after transfection, its levels were always lower in UBR5^{-/-} during RIG-I 2CARDs and MAVS overexpression. Surprisingly, TBK1-induced RIG-I expression was no different between WT and UBR5^{-/-} cells at 24 hrs (**Extended Data Fig.6b**). TBK1 activates IRF3/7, IFN-I expression and subsequent JAK-STAT1/2 pathway to amplify RIG-I expression, while RIG-I/MAVS overexpression activates more complex signaling events than just TBK1, for example, NF-κB and MAPK etc. Therefore, our results imply that UBR5

Penghua Wang, Ph.D., Assoc. Prof.
Department of Immunology, School of Medicine
The University of Connecticut Health Center
263 Farmington Ave, Farmington, CT 06030
Email: pewang@uchc.edu
Tel: 860-679-6393

activity requires upregulation by RLR-MAVS signaling, providing positive feedback to UBR5-mediated RLR transcription. Consistent with the TBK1 overexpression, recombinant human IFN- β -induced expression of RLR and other conventional ISG was no different between WT and *UBR5*^{-/-} cells (See the previous Response 1.1 and discussion in Page 17, Line 421-435).

We described and discussed these new results (Page 9-10, Line 230-231; Page 17-18, Line 435-441).

Reviewer 3 (Remarks to the Author):

I am satisfied with the authors comments and changes of the manuscript on addressing my concerns.

Response: We thank this reviewer for the positive feedback to our revisions.

Penghua Wang, Ph.D., Assoc. Prof.
Department of Immunology, School of Medicine
The University of Connecticut Health Center
263 Farmington Ave, Farmington, CT 06030
Email: pewang@uchc.edu
Tel: 860-679-6393

Point-by-point response to the reviewers' comments

REVIEWERS' COMMENTS

Response: We thank the reviewers for the positive feedback to our revisions.